# T³-S2S: Training-free Triplet Tuning for Sketch to Scene Synthesis in Controllable Concept Art Generation

**Zhenhong Sun**[1]*     *zhenhong.sun@anu.edu.au*
**Yifu Wang**[2]†     *1fwang927@gmail.com*
**Yonhon Ng**[3]     *yonhon.ng@hotmail.com*
**Yongzhi Xu**[3]     *z5105843@zmail.unsw.edu.au*
**Daoyi Dong**[4]     *daoyi.dong@uts.edu.au*
**Hongdong Li**[1]     *hongdong.li@anu.edu.au*
**Pan Ji**[2]     *peterji530@gmail.com*

[1] *Australian National University*
[2] *Vertex Lab*
[3] *Independent Researcher*
[4] *University of Technology Sydney*

**Reviewed on OpenReview:** *https://openreview.net/forum?id=lyn2BgKQ8F*

## Abstract

2D concept art generation for 3D scenes is a crucial yet challenging task in computer graphics, as creating natural intuitive environments still demands extensive manual effort in concept design. While generative AI has simplified 2D concept design via text-to-image synthesis, it struggles with complex multi-instance scenes and offers limited support for structured terrain layout. In this paper, we propose a **T**raining-free **T**riplet **T**uning for **S**ketch-to-**S**cene (**T³-S2S**) generation after reviewing the entire cross-attention mechanism. This scheme revitalizes the ControlNet model for detailed multi-instance generation via three key modules: **Prompt Balance** ensures keyword representation and minimizes the risk of missing critical instances; **Characteristic Priority** emphasizes sketch-based features by highlighting TopK indices in feature channels; and **Dense Tuning** refines contour details within instance-related regions of the attention map. Leveraging the controllability of **T³-S2S**, we also introduce a feature-sharing strategy with dual prompt sets to generate layer-aware isometric and terrain-view representations for the terrain layout. Experiments show that our sketch-to-scene workflow consistently produces multi-instance 2D scenes with details aligned with input prompts. Code is available at the official repository https://github.com/Tencent/Triplet_Tuning, with a maintained mirror for future updates at https://github.com/EngineeringAI-LAB/triplet_tuning.

## 1 Introduction

Scene generation plays a significant role in visual content creation across various domains, including video gaming, animation, filmmaking, and virtual/augmented reality. Traditional methods heavily rely on manual efforts, which require designers to transform initial sketches into detailed multi-instance scene concept art through numerous iterations. Recently, technological innovations such as Stable Diffusion (Rombach et al., 2022; Podell et al., 2023) equipped with ControlNet (Zhang et al., 2023b) and integrated with advanced text-to-image technologies (Kim et al., 2023), have streamlined this process. These advancements have notably decreased the workload for designers by automating the conversion of simple sketches into complex scenes. While these technologies perform well with common scenes involving typical instances, they struggle with

---

*Work conducted with Yifu Wang, Yonhon Ng, Yongzhi Xu, and Pan Ji during the author's internship at Tencent in 2024.
†Correspondence to: Yifu Wang <1fwang927@gmail.com>.

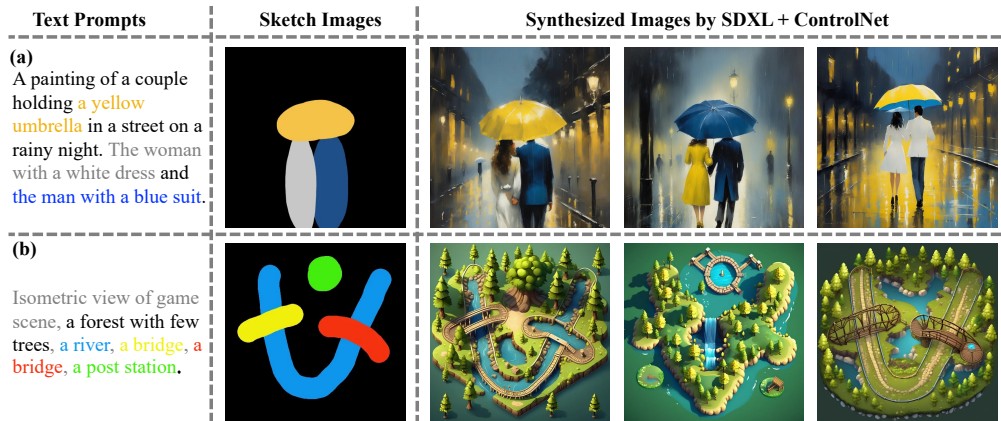

Figure 1: The SDXL-base model (Podell et al., 2023) and ControlNet model (Xinsir, 2023) perform well with common instances like humans, but they struggle with complex multi-instance scenes involving small instances and fail to accurately follow users' prompt.

generating complex multi-instance scenes and often miss fine details. For the terrain layout, designers still manually interpret concept art and build terrain by dragging grids with preset tools.

Alternatively, sketch-based synthesis can adopt multi-instance strategies by incorporating instance layouts via bounding boxes to guide the generation of multiple elements. While effective, most existing methods (Yang et al., 2023; Li et al., 2023; Liu et al., 2023; Sun et al., 2024; Wang et al., 2024b; Zhou et al., 2024) are training-based and require adaptation for sketches, relying on large datasets often restricted by copyright in gaming, animation, and film. In contrast, training-free approaches (Xie et al., 2023; Chen et al., 2024; Feng et al., 2022; Kim et al., 2023) leverage attention maps to exploit inherent model capabilities, offering flexible adaptability and low cost for new tasks. Building on this, we retain ControlNet's sketch-following ability with a training-free tuning mechanism requiring no additional data. Through a theoretical cross-attention analysis, we find that imbalanced prompt energy and value non-prominence undermine the competitiveness of instances and increase coupling among similar ones, leading to deviation from intended prompts.

In this paper, we introduce a **T**raining-free **T**riplet **T**uning for **S**ketch-to-**S**cene (**T**[3]-**S2S**) generation via three modules. Prompt balance adjusts instance-specific keyword energy to ensure all instances remain competitive. Characteristics priority amplifies instance-specific traits using a TopK selection from value matrices to enhance channel prominence in the feature map. Dense tuning adapted from (Kim et al., 2023) strengthens instance-related contour details in the attention map of the ControlNet branch. Together, these three form a unified triplet strategy that generates detailed, multi-instance 2D scenes that closely align with input prompts and sketches. Based on the controllability, we propose a twin structural concept framework that uses a dual-branch prompt synthesis with a mask-guided feature-sharing mechanism to generate layer-aware isometric and terrain-view representations for reconstructing the terrain layout. Experimental evaluations indicate that our T[3]-S2S approach boosts the performance of existing text-to-image models, consistently producing detailed, multi-instance scenes that closely align with the input sketches and input prompts.

The key contributions of our work are summarized as follows:

- We theoretically investigate the underlying cross-attention mechanisms and identify the imbalance of prompt energy and value non-prominence, leading to deviation from intended prompts.
- The triplet tuning advances controllable concept art generation by balancing token competition, enriching attention expression, and accentuating each instance's characteristics.
- Our **T**[3]-**S2S** workflow consistently generates detailed multi-instance 2D images and terrain layout aligned with input prompts.

## 2 Related Works

**Diffusion Techniques**. Recently, diffusion models (Ho et al., 2020; Rombach et al., 2022; Podell et al., 2023; Saharia et al., 2022) have marked a major breakthrough, improving the fidelity and realism in text-to-image generation, which shows infinite potential for concept art generation. The emergence of diffusion models has also advanced the development of 3D content generation tools. However, creating high-fidelity 3D scenes from images remains a complex task due to the diversity and intricacy of object shapes and appearances.

**Sketch-to-image Synthesis**. While text-to-image models can generate high-fidelity, realistic images, they struggle to accurately convey complex layouts with text prompts alone. In the field of diffusion-based generation, notable works include ControlNet (Zhang et al., 2023b), Make-a-scene (Gafni et al., 2022), and T2I Adapter (Mou et al., 2023) handle various additional visual conditions, including sketches, while methods like Dense Diffusion (Kim et al., 2023), SpaText (Avrahami et al., 2023) and MultiDiffusion (Bar-Tal et al., 2023) focus specifically on sketch-based inputs. In particular, Dense Diffusion is a training-free approach that adjusts the attention map by amplifying sketch-relevant tokens and downplaying less important ones, allowing the model to better distinguish between instances. ControlNet is a powerful solution for sketch-to-scene generation, recognized for its exceptional ability to accurately follow conditions. However, these models often struggle with complex multi-instance scene generations, particularly when handling unusual or unique instances, and frequently overlook smaller instances. Recently, Xu et al. (2024) proposed an efficient pipeline for automatically generating interactive 3D game scenes from users' natural input sketches using the SDXL and ControlNet models. However, the approach is also limited by the diversity and multi-instance representation in the intermediate 2D isometric image generation.

**Multi-instance Synthesis**. Multi-instance synthesis is closely related to sketch-to-scene generation due to its controllable layout. Training-free modulations (Xie et al., 2023; Chen et al., 2024; Lian et al., 2023; Feng et al., 2022) and training-based fine-tuning methods (Yang et al., 2023; Li et al., 2023; Liu et al., 2023; Sun et al., 2024; Wang et al., 2024b; Zhou et al., 2024) tackle the challenge of diffusion models accurately representing multiple instances with bounding boxes. For example, Li et al. (2023) (GLIGEN) used bounding box coordinates as grounding tokens, integrating them into a gated self-attention mechanism to improve positioning accuracy, while Liu et al. (2023) employed a latent object detection model to separate objects, masking conflicting prompts and enhancing relevant ones. Despite existing methods of generating images with correct positions, these box-based approaches struggle with simple sketch inputs and fail to strictly follow the designer's sketch. Our work leverages ControlNet's sketch-following capabilities and investigates the challenges of synthesizing multiple instances. We aim to design a training-free tuning mechanism to enhance modeling within cross-attention, addressing these challenges effectively.

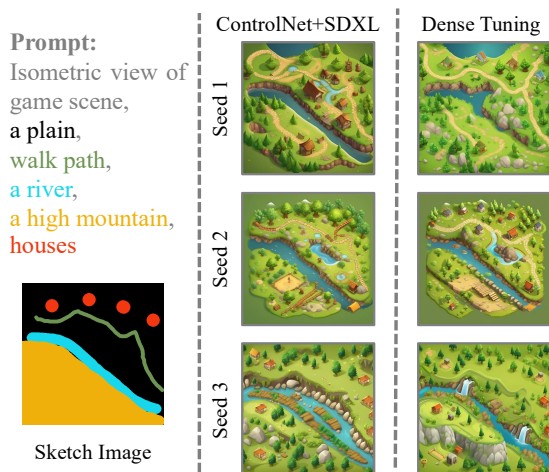

Figure 2: The SDXL-base (Podell et al., 2023) and ControlNet models (Xinsir, 2023) struggle with complex multi-instance scenes based on sketch images and text prompt, even with improved dense tuning (Kim et al., 2023).

## 3 Controllability Analysis of Cross-attention Mechanism

**Problem Statement.** ControlNet (Zhang et al., 2023a) is adopted alongside SDXL (Podell et al., 2023) as our baseline for concept art generation due to its controllability in sketch-to-image generation. Given a textual prompt $\mathbf{c}_g = \{c^i\}_{i=0}^{l}$ ($l$ is words number) and a sketch image $\mathbf{C}_s \in \mathbb{R}^{h \times w}$, the system generates images via cross-attention mechanism in the UNet (Rombach et al., 2022), aligning spatial features with text

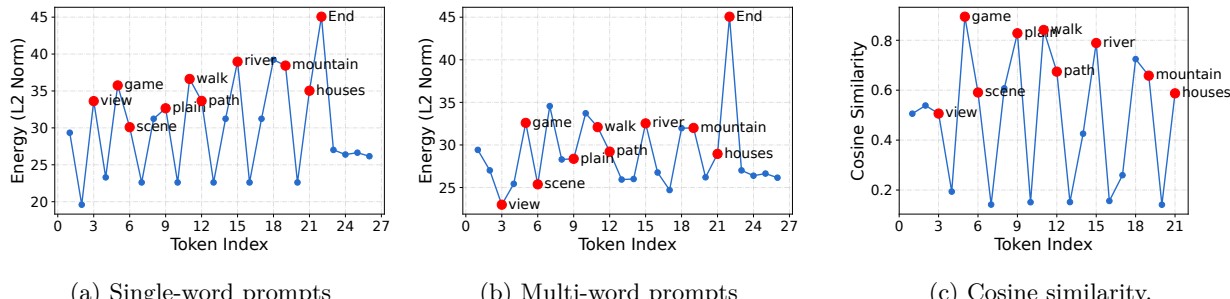

(a) Single-word prompts      (b) Multi-word prompts      (c) Cosine similarity.

Figure 3: Embedding energy comparison between a global prompt ("Isometric view of game scene, a plain, walk path, a river, a high mountain, houses.") and single-word prompts (each keyword separated and embedded individually, higher than that in the group). The energy imbalance can lead to attention competition, and low-energy small instances ("path" and "houses") are easily forgotten. (c) Cosine similarity between embeddings of (a) and (b)

embeddings $\mathbf{S} \in \mathbb{R}^{n \times d}$ (encoded from $\mathbf{c}_g$):

$$\mathbf{F}_m = \mathbf{A}_m \mathbf{V}_m = \mathrm{softmax}\left(\mathbf{Q}_m \mathbf{K}_m^\top / \sqrt{d_m}\right) \mathbf{V}_m,$$

where $m$ is layer number, $n$ the number of text tokens, and $d$ the token embedding dimension. $\mathbf{Q}_m \in \mathbb{R}^{b_m \times d_m}$ comes from fore-features, and $\mathbf{K}_m, \mathbf{V}_m \in \mathbb{R}^{n \times d_m}$ are derived from text embeddings $\mathbf{S}$ by linear projections, where $b_m$ is the flattened spatial dimension, and $d_m$ is the embedding dimension. As shown in Figure 2, the system struggles with complex multi-instance natural scenes guided by sketch images and text prompts, often failing to render all intended objects.

Dense Diffusion (Kim et al., 2023) can effectively highlight different instances' attention values based on the sketches in the attention map $\mathbf{A}_m \in \mathbb{R}^{b_m \times n}$, but directly applying its strategy to ControlNet leads to performance degradation. To adapt it, we make an improved **Dense Tuning (DT)** (Details in **Appendix B**): 1) Only add the dense tuning at the "down_block_2" and "mid_block_0" of ControlNet; 2) Only use the expand function, not use the suppress function in the self-attention. Despite this, the image quality improves, but instances with small areas, such as "path" and "houses", are easily neglected in the final image, despite having strong responses in feature maps. This highlights that beyond tuning attention maps alone, there remains room to explore controllability within the broader cross-attention process. In particular, the roles of $\mathbf{K}_m$ in shaping attention distributions and $\mathbf{V}_m$ in contributing to the final feature outputs have received limited attention.

**Imbalance of Prompt Energy.** In practice, increasing prompt weights (e.g., "(houses:1.5)" in WebUI) can enhance the visibility of specific instances in multi-instance generation. To understand the underlying mechanism, we refer to Henry et al. (2020) to decompose each token embedding from $\mathbf{S} = [\mathbf{s}^1, \ldots, \mathbf{s}^n] \in \mathbb{R}^{n \times d}$ as:

$$\mathbf{s}^i = E^i \cdot \hat{\mathbf{s}}^i, \quad \text{where } E^i = \|\mathbf{s}^i\|, \ \|\hat{\mathbf{s}}^i\| = 1,$$

where, $E_i$ represents the prompt energy (L2 norm), and $\hat{\mathbf{s}}^i$ denotes the normalized embedding. Since both keys and values in cross-attention are linear projections of $\mathbf{s}^i$, their magnitudes are bounded:

$$\|\mathbf{k}^i\| = \|\mathbf{s}^i \mathbf{W}_K\| \leq E^i \cdot \|\mathbf{W}_K\|, \quad \|\mathbf{v}^i\| = \|\mathbf{s}^i \mathbf{W}_V\| \leq E^i \cdot \|\mathbf{W}_V\|.$$

Thus, tokens with lower energy yield weaker keys $\mathbf{k}^i$ and values $\mathbf{v}^i$, diminishing their impact on attention and feature aggregation.

Considering the prompts in Fig. 2, we analyze the embedding energy within a global prompt and each keyword embedding separately, as shown in Fig. 3. Global encoding obviously lowers the energy of the individual tokens, especially the instances like "path" and "houses", aligns with the observed instance omission in Fig. 2. The final attention value $a$ of the valid token can be represented as $a \approx E\bar{a}$, where $\bar{a}$ denotes the

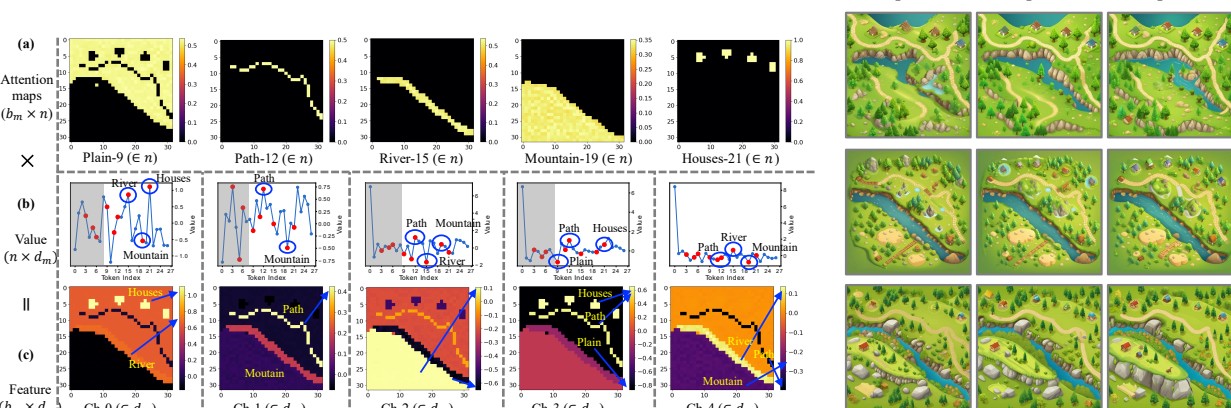

Figure 4: Interaction between attention maps and value matrices with prompts from Fig. 2 using dense tuning. (a) Attention maps highlight strong sketch relevance. (b/c) Five-channel value-feature pairs reveal the importance of extrema. Despite feature enhancement improving instance chances, forgetting still occurs as extrema are not prominent. Statistics are shown in Fig. 10 and **Appendix C**.

Figure 5: Generations by amplifying the TopK extrema twice in the value matrices based on the pipeline in Fig. 4, where most instances appear, but uniform amplification also introduces noise.

mean attention weight, empirically observed as approximately 0.43 across 100 prompt sets. After applying exponentiation, the ratio between two attention values becomes:

$$Ratio = e^{E^2 \bar{a}} / e^{E^1 \bar{a}} = e^{(E^2 - E^1)\bar{a}}.$$

Due to exponential weighting, energy differences of $\{1, 3, 5\}$ result in an attention *Ratio* disparity of $\{1.54, 3.63, 8.58\}$ before the softmax average, and higher-energy tokens easily win the attention competition. Even if a low-energy token receives high attention via tuning, its contribution remains limited due to the low magnitude of its value vector $\mathbf{v}_i$ which weakens the aggregated feature in the product $\mathbf{A}_m \mathbf{V}_m$. This energy imbalance, which allows dominant tokens to overshadow weaker ones and increases the risk of instance omission in multi-instance synthesis, underscores the importance of balancing and scaling prompt energy as an interesting perspective to improve multi-instance scene generation.

**Non-prominence of Value Matrices.** As a core part of cross-attention, the interaction between attention maps and value matrices shapes each channel's characteristics to instance-level patterns such as geometry and attributes. To maximize expressiveness, each channel should focus on distinct semantics, similar to SENet's calibration (Hu et al., 2018). The aggregated feature at channel $j$ is:

$$\mathbf{f}_m^j = \mathbf{A_m} \cdot \mathbf{v}_m^j = \{[a_m^{z,1},\ a_m^{z,i},\ \ldots,\ a_m^{z,n}] \cdot [v_m^{1,j},\ v_m^{i,j},\ \ldots,\ v_m^{n,j}]\}_{z=1}^{b_m}.$$

As shown in Fig. 4(a), dense tuning sharpens attention, allowing each spatial position $z$ to attend to a single instance via high $a_m^{z,i}$. However, The non-prominence between value values can cause responses to multiple instances, leading to entangled features. Specifically, the extreme values determine which instances are activated spatially, consistent with the observations in Fig. 4(b) and (c).

To probe this, we double the TopK values in each channel of the value matrices, as shown in Figure 5. As $K$ increases, the model initially synthetizes all instances (e.g., at $K = 2$), but it also introduces excessive noise (e.g., over-detailed houses). This demonstrates that stronger value prominence improves token competitiveness, but requires a trade-off between instance completeness and visual clarity via controlled value amplification.

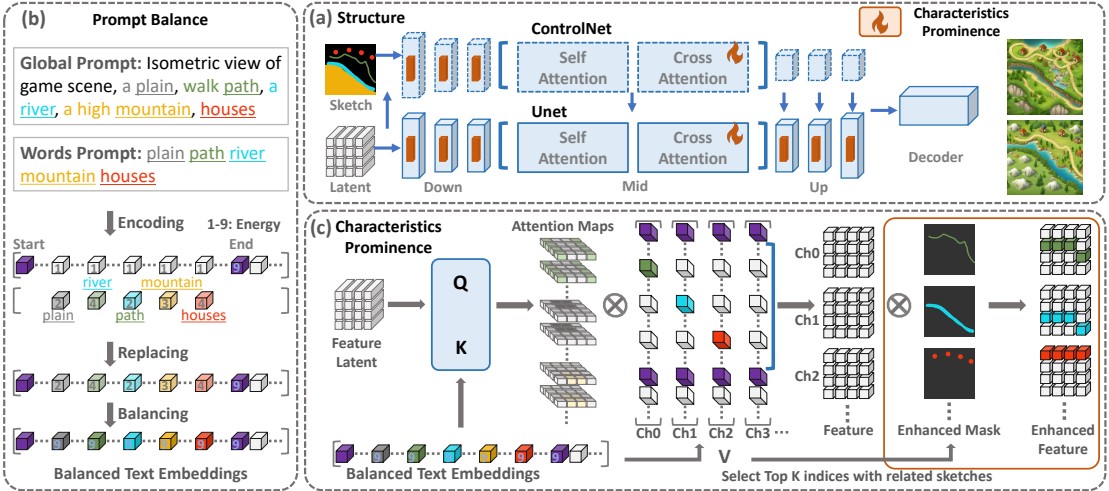

Figure 6: Overview of the proposed training-free triplet tuning strategy in the frozen pre-trained latent diffusion model. (a) The orange parts indicate the proposed module plugged into the ControlNet and U-Net framework. (b) The left part shows the energy tuning of the prompt balance. (c) The bottom part indicates the training-free tuning of the characteristics priority. $\mathbf{T}^3$-$\mathbf{S2S}$ ensures the final generation responds effectively to both text and sketch inputs.

## 4 Diffusion for Controllable Concept Art Generation

To address the cross-attention challenges discussed in Section 3, we propose a training-free triplet tuning strategy for controllable concept art generation, as illustrated in Fig. 6. This strategy consists of three modules: **Prompt Balance (PB)**, **Characteristics Priority (CP)**, and **Dense Tuning (DT)**:

(1) **Prompt Balance**: This module identifies instance keywords within global text prompts, replaces their embeddings with corresponding single-word embeddings, and adjusts the energy of these keyword embeddings to maintain balance. By balancing the energy of the keyword embeddings, the method enhances the representation of instances within key and value matrices. This process improves the competitiveness of instance tokens among all tokens, ensures consistency across instance tokens, and reduces the likelihood of overlooking rare or unusual instances.

(2) **Characteristics Priority**: This module selects instance-related tokens and their sketches by identifying the TopK values for each channel in the value matrices, creating an instance-specific mask. The mask is then used to scale up the feature map for the corresponding channel. This approach enhances the distinction of instances within the multi-channel feature space without additional parameters, ensuring that instances' characteristics are more prominently emphasized.

(3) **Dense Tuning**: While prompt balance increases the strength of the embedding matrices related to instances, enhancing their competitiveness in the attention map, the overall strength of the attention map remains suboptimal. Meanwhile, given that more contour information resides in the ControlNet branch, we employ dense modulation directly within this branch to augment the attention map for better modulation. Since attention manipulation has been extensively explored in prior works (Hertz et al., 2022; Xie et al., 2023; Kim et al., 2023; Chen et al., 2024), we only adopt a modified dense tuning scheme from Kim et al. (2023), detailed in Section 3.

In the subsequent section, we will provide a detailed explanation of two newly designed modules and their underlying rationales.

### 4.1 Prompt Balance

As discussed previously, the imbalance of prompt energy influences competition among key and value matrices related to instances, increasing the risk of missing instances. To mitigate this, we propose a plug-in prompt balance strategy for the text embeddings that enhances energy uniformity across keywords and scales up

values, as displayed in Figure 6 (b). Specifically, we use an NLP network (e.g., the SpaCy library) to identify instance keywords from the global text prompts $\mathbf{c}_g = \{c^i\}_{i=0}^l$, resulting in reorganized instance keyword prompts $\{c^i\}^{\mathbf{q}}$, where $\mathbf{q}$ is the **indices vector** of keywords in $\mathbf{c}_g$. Then, we encode both the global text prompts and each instance keyword prompts separately into text embeddings $\mathbf{S}_g = \{\mathbf{s}_g^i\} \in \mathbb{R}^{n \times d}$ and $\{\mathbf{s}_w^i \in \mathbb{R}^{1 \times d}\}^{\mathbf{q}}$ by a text encoding network. Next, we replace the embedding of keywords in $\mathbf{S}_g$ with the single-word embedding of $\mathbf{S}_w$ to form a new combined embedding $\mathbf{S}_r \in \mathbb{R}^{n \times d}$:

$$\mathbf{S}_r = \{\mathbf{s}_r^i\} = \{\mathbf{s}_w^i\}, \quad \text{if } y \in \mathbf{q}, \quad \text{otherwise } \{\mathbf{s}_g^i\}.$$

Generally, the special "end of text" token (located $i_{\text{end}}$) always has the maximum energy as shown in Fig. 3, which could be the upper bound for us to scale up the embeddings of the keywords in $\mathbf{S}_r$ that all keywords have balanced energy relative to the "end of text" token embedding, mathematically represented as:

$$\{\mathbf{s}_r^i\}^{\mathbf{q}} = \{(E_r^{i_{\text{end}}}/E_r^i) \cdot \mathbf{s}_w^i\}^{\mathbf{q}}, \quad \text{where} \quad E_r^{i_{\text{end}}} = \|\mathbf{s}_r^{i_{\text{end}}}\| \text{ and } E_r^i = \|\mathbf{s}_r^i\|.$$

Finally, the balanced text embeddings, denoted as $\mathbf{S}_b$, enhance the instance-based token values in the key, value matrices, and attention map, improving their competitiveness and consistency for more concise and effective instance representation.

## 4.2 Characteristics Priority

While balanced text embeddings help equalize instance competition, value matrices still face low prominence. As discussed in Section 3, enhancing TopK values per channel improves prominence. To balance instance completeness and noise clarity, we propose a CP technique that applies localized TopK enhancement to specific sketch regions for features. Specifically, instead of directly enhancing the TopK values along the $n$ dimension in the value matrix $\mathbf{V}_m \in \mathbb{R}^{n \times d_m}$, we apply enhancement based on the indices of the TopK values on the feature map $\mathbf{F}_m \in \mathbb{R}^{b_m \times d_m}$ (before the residual adding). For each channel in the value matrix $\mathbf{V}_m$, we find the indices of the TopK values across all valid tokens (between "start" and "end" tokens):

$$\mathbf{Y}_K = \{\mathbf{y}_K^j\}_{j=0}^{d_m} = \text{TopK}(\text{abs}(\mathbf{V}_m[1:i_{end}]), K) \in \mathbb{R}^{K \times d_m},$$

where $K$ is the number of top values considered. For $j$th channel $\mathbf{f}_m^j \in \mathbb{R}^{b_m}$ in $\mathbf{F}_m$, we check whether each index $i$ in $\mathbf{y}_K^j$ belongs to the instance keyword vector $\mathbf{q}$. If it does, the index $i$ corresponds to a specific instance token $i \in \mathbf{q}$. Then the sketch $\mathbf{u}_m^i \in \mathbb{R}^{b_m}$ of the instance at the current scale will be summed together to generate an enhancement mask $\mathbf{h}_m^j$ for the $j$th channel:

$$\mathbf{h}_m^j = \sum \mathbf{u}_m^i, \quad \text{if } i \in \{\mathbf{y}_K^j \text{ and } \mathbf{q}\}.$$

The whole mask matrices $\mathbf{H}_m = \{\mathbf{h}_m^j\}_{j=0}^{d_m} \in \mathbb{R}^{b_m \times d_m}$ are used to proportionally scale up the corresponding values in the feature map $\mathbf{F}_m$ by a factor $\beta$, obtaining the enhanced feature map $\hat{\mathbf{F}}_m$:

$$\hat{\mathbf{F}}_m = \mathbf{F}_m + \beta \cdot \mathbf{H}_m \odot \mathbf{F}_m,$$

where $\odot$ denotes element-wise multiplication. This enhancement emphasizes instance tokens in the multi-channel feature space, improving instance distinction. The CP technique strengthens the attention mechanism by amplifying instance-relevant regions in the feature map, even with small sketches.

## 4.3 Structural Concept Representations

Building on the previous modules, the $\mathbf{T}^3$-**S2S** can produce detailed, multi-instance scenes that align closely with input prompts and sketches. To further enhance scene structure, especially in maintaining terrain consistency while separating foreground objects, we propose a twin structural concept representations framework. As depicted in Figure 7, it adopts a dual-branch prompt synthesis with mask-guided feature-sharing mechanism: one for the complete isometric image with full sketches and another for the empty terrain with non-foreground sketches.

The feature maps for the complete isometric image and the empty terrain are denoted as $\mathbf{F}_{\text{iso}}(t)$ and $\mathbf{F}_{\text{terr}}(t)$, respectively, at inference step $t$. The terrain branch updates with selective and progressive blending, defined as:

$$\hat{\mathbf{F}}_{\text{ter}} = \mathbf{F}_{\text{ter}} + \mathbf{M} \odot \gamma(t) \cdot \left(\mathbf{F}_{\text{iso}} - \mathbf{F}_{\text{ter}}\right), \quad \gamma(t) = \gamma_0 \cdot \left(\frac{t}{T}\right)^{\alpha},$$

where $\odot$ denotes element-wise multiplication, $\gamma_0$ the initial scaling factor. $\mathbf{M}$ is a binary mask derived from the sketch sub-prompts, with 1 indicating background regions and 0 indicating foreground regions (e.g., houses, bridges). The dynamic scaling $\gamma(t)$ ensures that the terrain branch inherits global features from the isometric branch at the early stages, while allowing independent refinement later, with the mask $\mathbf{M}$ preserving foreground integrity. Together, mask-guided sharing and dynamic scaling enable harmonious terrain generation and effective foreground-background separation.

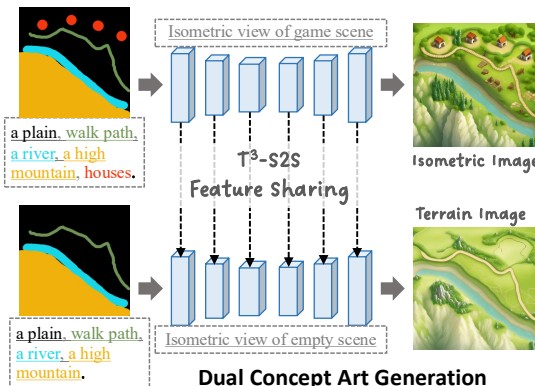

Figure 7: Twin-structure framework with dual-branch prompt synthesis, where a mask-guided feature-sharing mechanism handles full-sketch isometric and non-foreground terrain images.

## 5 Experiments

### 5.1 Implementation Details

**Baselines.** Our baseline leverages the sketch-processing capabilities of the *ControlNet* model (Zhang et al., 2023b; Xinsir, 2023) with the *SDXL-base* model (Podell et al., 2023), compared with two SDXL-base sketch-oriented approaches: the training-based *T2I Adapter* (Mou et al., 2023) and the training-free *Dense Diffusion* (Kim et al., 2023). We also further validate PB with Attend-and-Excite (Chefer et al., 2023) (**Appendix G**), and T³-S2S with T2I adapter (**Appendix H**).

**Setup.** In the triplet tuning scheme, the prompt balance module is integrated into the text encoding process, while the characteristics priority module are incorporated across all cross-attention layers. The dense tuning module is specifically added to the "down_blocks 2" layers and the "mid_blocks 0" layers within the ControlNet branch. We set $K = 2$, $\alpha = 5$, $\gamma_0 = 1$ and $\beta = 1$. During inference, we use the default Euler Discrete Scheduler (Karras et al., 2022) with 32 steps and a guidance scale of 9 at a resolution of $1024 \times 1024$. All experiments are conducted on a single Nvidia Tesla V100 GPU.

**Metrics.** Given that our current approach involves sketch-based multi-instance scene generation, existing benchmarks should be adjusted for our evaluation, such as adding sketch inputs for T2I-CompBench (Huang et al., 2023). Therefore, we design 50 complex sketch scenes, each with more than four sub-prompts, encompassing various terrains (plains, mountains, deserts, tundra, cities) and diverse instances (rivers, bridges, stones, castles). We utilize CLIP-Score (Hessel et al., 2021) for the global prompt and image, and evaluate the CLIP-Score for each background prompt and instance prompts by cropping the corresponding regions. Additionally, we conduct a user study to assess different variants of our approach, using a 1-5 rating scale to evaluate image quality, placement, and prompt-image consistency. Details can be found in **Appendix D&F**.

### 5.2 Main Results

**Qualitative Evaluation.** Building upon the scene design, we show two representative and complex multi-instance scene scenarios, each incorporating a diverse array of elements to foster varied interactions. We evaluate several approaches, with visual comparisons displayed in Figure 8. Due to the specialized nature of this task, most existing solutions will overlook objects aligned with the prompts. When combined with the triplet tuning strategy, our T³-S2S method improves the generation performance of existing SDXL models. For example, Figure 8 (a) showcases the enhanced detail in smaller instances such as "houses" and "path", and even less common elements like "mountains". Similarly, Figure 8 (b) illustrates the effective

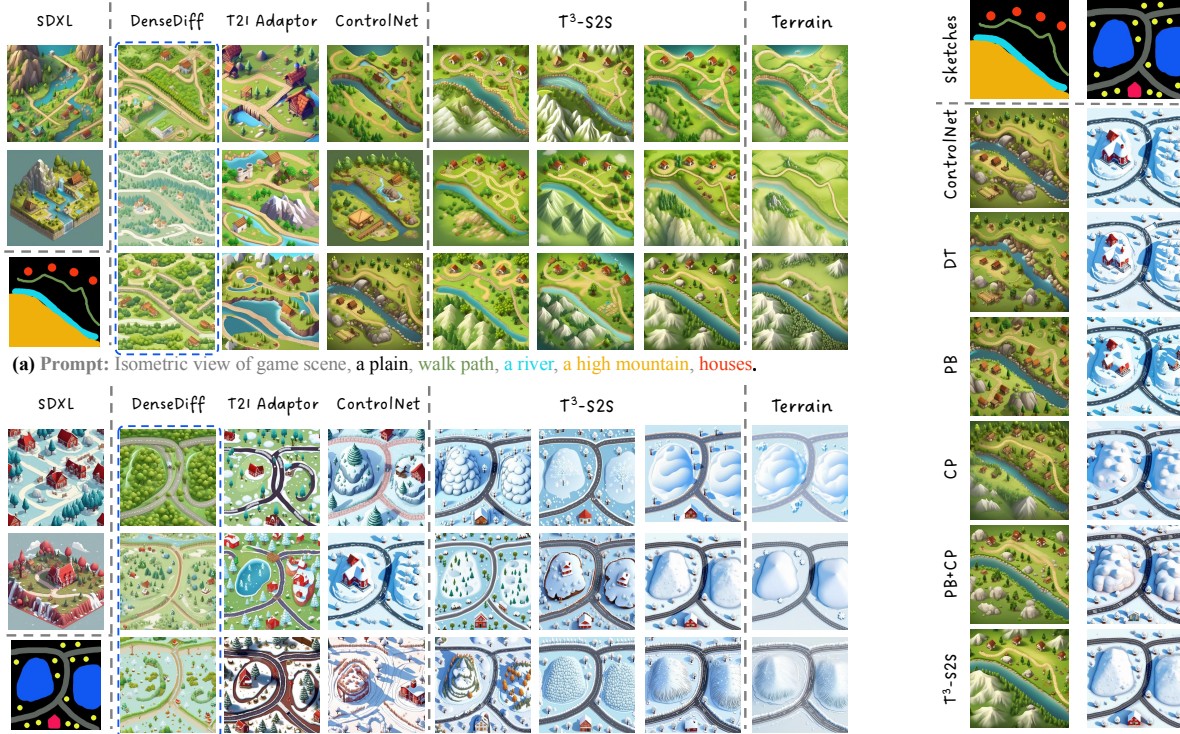

**(a) Prompt:** Isometric view of game scene, **a plain**, walk path, a river, a high mountain, houses.

**(b) Prompt:** Isometric view of game scene, **a field with ice and snow**, iced hills, winding road, trees, a red house.

Figure 8: Qualitative comparison with baseline methods. (a) T³-S2S performs well for smaller instances like "houses" and "path" , and unusual "mountain". (b) T³-S2S performs well with a large number of small instances "trees". Note that the original Dense Diffusion (Kim et al., 2023) based on SD V1.5 (Rombach et al., 2022), has limited prompt response capabilities. For a fair comparison, we apply it to the SDXL model.

Figure 9: Qualitative comparison of different inserted modules based on the ControlNet and SDXL model. The quantitative results are present in Table 2.

Table 1: Evaluation results on T2I-CompBench. # indicates the method evaluated in the new benchmark, others from the T2i-CompBench paper.

| Model | Color B-VQA↑ | 2D Spatial UniDet↑ | Numeracy UniDet↑ |
|---|---|---|---|
| SDXL | 0.5879 | 0.2133 | 0.4991 |
| ControlNet# | 0.5925 | 0.2686 | 0.4639 |
| ControlNet+DT# | 0.6154 | 0.3419 | 0.5143 |
| **T³-S2S#** | 0.6872 | 0.4256 | 0.6257 |
| Stable v3 | 0.8132 | 0.3200 | 0.6174 |
| FLUX.1 | 0.7407 | 0.2863 | 0.6185 |

Table 2: Comparison of CLIP-Score across several variants based on SDXL model, evaluated on whole images, masked instance regions, and masked background regions. Includes user study ratings on a scale of 1–5.

| Model | Global↑ | Instances↑ | Background↑ | User↑ |
|---|---|---|---|---|
| DenseDiff | 0.3438 | 0.2459 | 0.2543 | 2.25 |
| T2I-Adaptor | 0.3422 | 0.2405 | 0.2523 | 2.17 |
| ControlNet | 0.3435 | 0.2427 | 0.2531 | 2.38 |
| +PB | 0.3439 | 0.2472 | 0.2559 | 2.62 |
| +DT | 0.3430 | 0.2469 | 0.2547 | 3.18 |
| +CP | 0.3457 | 0.2503 | 0.2565 | 3.40 |
| +PB+CP | 0.3466 | 0.2552 | 0.2571 | 3.55 |
| **T³-S2S** | **0.3491** | **0.2566** | **0.2581** | **3.79** |

generation of numerous small instances, like "trees". By leveraging the triplet tuning strategy within the cross-attention mechanism, our approach consistently generates detailed, multi-instance scenes that closely adhere to the original sketches and texts. Additional game and common scenes are provided in Fig. 15 and **Appendix D&E**.

**Quantitative T2I-CompBench.** We evaluate common scene generation on the large-scale T2I-CompBench (Huang et al., 2023) using Color (B-VQA), 2D Spatial (UniDet), and Numeracy (UniDet) metrics (Standard: 300 prompts/metric, 10 images/prompt). As our method requires sketch inputs, we use

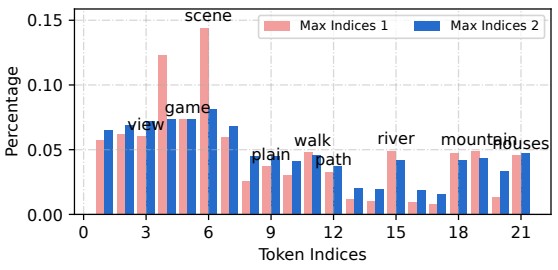

(a) The first and second extreme points.

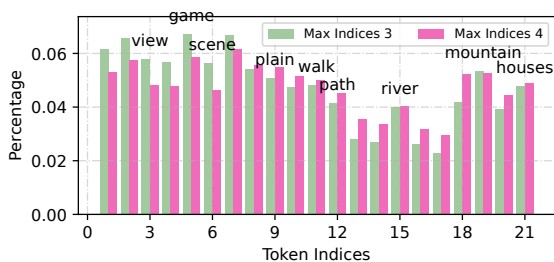

(b) The third and fourth extreme points.

Figure 10: Histogram of the distribution of extrema in Value matrices. At the first and second extrema, the probability of later instance tokens is lower than that of earlier tokens, so increasing their values helps enhance their representations. However, at the third and fourth extremes, the probabilities tend to converge, implying that only a few tokens are key to defining features.

GPT-4V to generate square sketches (similar to bounding boxes) for all instances, with manually adjusted sizes, presented in Table 1. Our method improves about 9.5% in the Color metric and 15.7% in the 2D Spatial metric over the ControlNet baselines. Especially for the Numeracy metric, we ensure that the 100 examples contain 1-3 small instance sketches with a minimal resolution of (ensuring 1 pixel at the minimal feature level). Compared to the SDXL baseline, ControlNet performs 3.5% drop on the Numeracy metric featuring small instances, while DenseDiffusion improves 1.4%. Our approach shows noticeable 16.2% gains over ControlNet for the generation of small instances.

**Quantitative Customized Evaluation.** We compare CLIP-Scores for global image, instances, and background across different variants and the base ControlNet. A user study is also conducted with a 1-5 rating scale. As shown in Table 2, our approach demonstrates better performance on the 50 complex multi-instance scenes, with improved fidelity and precision in aligning with text prompts and sketch layouts. The PB module shows modest improvement, while the CP and DT modules provide comparable enhancements. Combining these components allows our $T^3$-S2S approach to achieve a well-balanced outcome.

### 5.3 Ablation Study

**Module Comparison.** We perform an ablation study (Fig. 9) to evaluate the individual and combined effects of three modules, as well as the quantitative results in Table 2: (1) DT constrains instance regions within sketches; (2) PB improves small object visibility (e.g., "houses") but may introduce noise; (3) CP sharpens features and suppresses noise. PB+CP addresses most issues, and combining all three ($\mathbf{T^3}$-$\mathbf{S2S}$) yields the best sketch-prompt alignment between generated scene images and their corresponding sketches and texts, aligning with the quantitative results present in Table 2.

### 5.4 Top $K$ Anaylsis

In the module of characteristic prominence, two hyperparameters, $K$ and $\beta$, require meticulous tuning. $K$ determines the indices of extreme values within the value matrices. For our analysis, we save these indices and construct a histogram, as depicted in Fig. 10. We observe that the probability of later instance tokens achieving the maximum and second maximum values is comparatively lower than that of earlier tokens. Thus, increasing the value of later instance tokens will be beneficial for their representations. However, at the third and fourth extremes, the probabilities tend to converge, indicating that not every token is essential for defining key characteristics. Increasing values for later instances at this point would introduce additional noise. Therefore, setting $K = 2$ is advisable based on the observed trends. For $\beta$, which enhances the characteristics of instances within the feature matrix, an initial increase is beneficial. Nonetheless, there is a critical threshold beyond which increases in $\beta$ begin to disrupt the distribution within the value matrices.

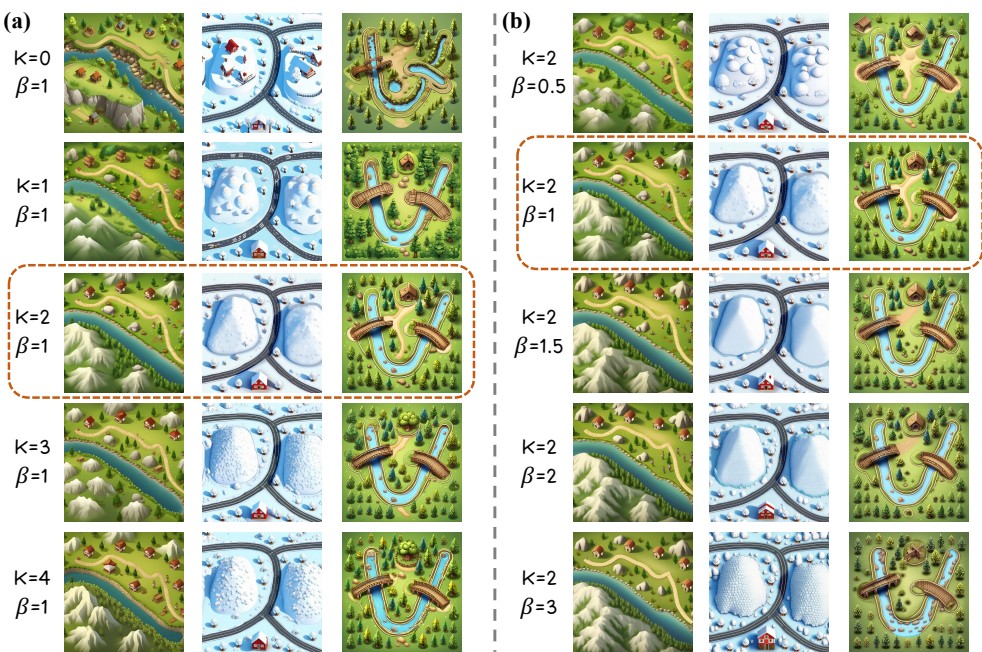

Figure 11: Visual comparison of two hyper-parameters $K$ and $\beta$ suggests that setting $K = 2$ and $\beta = 1$ is a favorable choice. With $\beta = 1$, increasing $K$ initially improves results but eventually adds noise. Fixing $K = 2$, higher $\beta$ values yield stable and favorable outcomes. Another comparison is presented in Figure 18.

**Hyper-parameter Comparison.** To validate our hypothesis, we study Top$K$ distribution in Fig. 10. In Fig. 10, later tokens have lower chances of reaching top values at the first and second extrema, so increasing their values helps enhance their representations. But at the third and fourth extrema, probabilities even out, implying only a few tokens are key to defining features. Based on this observation, we further examine the effects of varying $K$ and $\beta$ on generation quality in Fig. 11. When fixing $\beta = 1$, increasing $K$ initially improves visual quality by strengthening under-represented tokens, but further increasing $K$ introduces noticeable noise, as less relevant tokens are also amplified. Conversely, when fixing $K = 2$, larger $\beta$ values lead to more stable and consistently favorable results, indicating that moderate amplification of a small number of key tokens is sufficient to achieve effective feature modulation without disrupting overall coherence. These results suggest a trade-off between expressiveness and stability, where overly large $K$ increases noise, while appropriately scaled $\beta$ preserves structural consistency. Consequently, we identify $K = 2$ and $\beta = 1$ as a robust and effective configuration across our experiments.

# 6 Conclusion

In conclusion, our study on the training-free triplet tuning for sketch-to-scene generation has enhanced the ability of text-to-image models to process complex, multi-instance scenes. By incorporating prompt balance, characteristics prominence, and dense tuning, we have effectively addressed issues such as imbalanced prompt energy and value homogeneity, which previously resulted in the inadequate representation of unusual and small instances. Our experimental results confirmed that our approach not only preserves the fidelity of input sketches but also elevates the detail of the generated scenes. This advancement is vital in fields like video gaming, filmmaking, and virtual/augmented reality, where precise and dynamic visual content creation is crucial. Facilitating more efficient and less labor-intensive generation processes, our model offers a promising avenue for future developments in automated sketch-to-scene transformations.

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

## A    Discussion and Limitations

While our approach is innovative and enhances multi-instance scene generation, it also has some room to improve, primarily stemming from the inherent capabilities of the base model.

**Texture Generation.**  Generating fine-grained textures and detailed instances remains challenging. This issue is partly due to the limited ability of CLIP-based representations to precisely encode complex descriptive attributes. Furthermore, the characteristic prominence module prioritizes instance-level tokens, which can lead to reduced attention on adjective-level descriptors that are critical for texture and appearance modeling. For example, as illustrated in Fig. 17, in cases where multiple instances of the same object with different colors are specified, our method is generally able to generate the desired targets. However, attribute confusion may still occur, leading to mismatches between instances and their corresponding color attributes.

**Large Scene Challenge.**  Our method encounters difficulties in accurately generating expansive scenes (e.g., game maps exceeding $4096 \times 4096$ pixels). Processing such high resolutions directly incurs prohibitive computational costs, typically necessitating patch-based processing or upsampling strategies. However, a more fundamental constraint lies in the model architecture itself. Taking SDXL as an example, the network downsamples a $1024 \times 1024$ input to a minimum bottleneck feature size of $32 \times 32$. This imposes an intrinsic $32\times$ scaling limit, meaning the minimum controllable granularity for object generation is restricted to this ratio. For example, in the first case of Figure 16, the control over minute objects degrades at the bottleneck layer. Consequently, even with increased output resolution, accurately resolving and controlling tiny instances or intricate interactions below this structural floor remains a significant challenge.

**Concept Art Understanding.**  Additionally, the 3D scene shows that terrain and objects can be generated as concept art, but improvements are needed in understanding concept art, particularly in segmentation and depth estimation, which may not be accurately represented in the synthesized data. One potential remedy is to concurrently generate multiple modalities at the same time, such as RGB, semantic, depth, material, and object footprint, and fuse these results.

Building on our current achievements, we plan to further explore these areas in future work to improve detailed multi-instance sketch-to-scene 3D generation.

## B    Details of Dense Tuning

Dense Diffusion (Kim et al., 2023) can effectively amplify instance-specific attention responses in standard diffusion models. However, when adapting dense modulation strategies to ControlNet, it is crucial to account for the hierarchical roles of different layers. We follow the layer-and-step-wise empirical observations (Voynov et al., 2023; Wang et al., 2024a; Sun et al., 2024) that peripheral (early) layers predominantly encode low-level geometric cues and boundary information, while more central layers are responsible for shaping object contours and structural layouts. This behavior is consistent with the hierarchical design of diffusion UNets and ControlNet, where early steps preserve fine spatial details and later steps increasingly capture semantic structure. As demonstrated in Table 2, DT alone yields marginal quantitative improvement, indicating that DT primarily acts as a modulation mechanism rather than the main source of performance gain, which is instead driven by the proposed PB and CT modules. Thus, to evaluate its application, we introduce adapted variants of DT across different layers within our proposed PB+CT framework.

**Qualitative analysis.**  Figure 12 presents a qualitative comparison of Dense Tuning (DT) applied at different depths of ControlNet. Applying DT at `mid_block_0` already produces visually coherent and well-controlled results, while extending DT to `down_block_2 + mid_block_0` further improves structural consistency. In contrast, including earlier layers such as `down_block_1` introduces overly strong contour and edge responses, resulting in visually intrusive ControlNet outlines that degrade image naturalness. This observation indicates that early ControlNet layers predominantly encode low-level geometric cues, where dense modulation tends to over-amplify sketch boundaries rather than semantic structure.

**Efficiency analysis.** Table 3 reports GPU memory consumption and inference latency under different DT layer configurations. As DT is activated in additional layers, both memory usage and inference time increase

Table 3: Cost–benefit analysis of Dense Tuning (DT) applied to different layer combinations in ControlNet and U-Net within the PB+CT framework.

| ControlNet DT | U-Net DT | Memory (MB) | Inference Time (s) | Global↑ | Instances↑ | Background↑ |
|---|---|---|---|---|---|---|
| M0 | N/A | 18175 | 10.20 | 0.3453 | 0.2532 | 0.2548 |
| D2 + M0 | N/A | 18215 | 10.61 | **0.3491** | **0.2566** | **0.2581** |
| D1 + D2 + M0 | N/A | 18498 | 10.89 | 0.3429 | 0.2501 | 0.2522 |
| D2 + M0 | M0 | 18661 | 10.96 | 0.3494 | 0.2571 | 0.2569 |
| D2 + M0 | D2 + M0 + U0 | 18949 | 11.14 | 0.3411 | 0.2492 | 0.2552 |
| D2 + M0 | D1 + D2 + M0 + U0 + U1 | 19297 | 11.35 | 0.3401 | 0.2496 | 0.2547 |

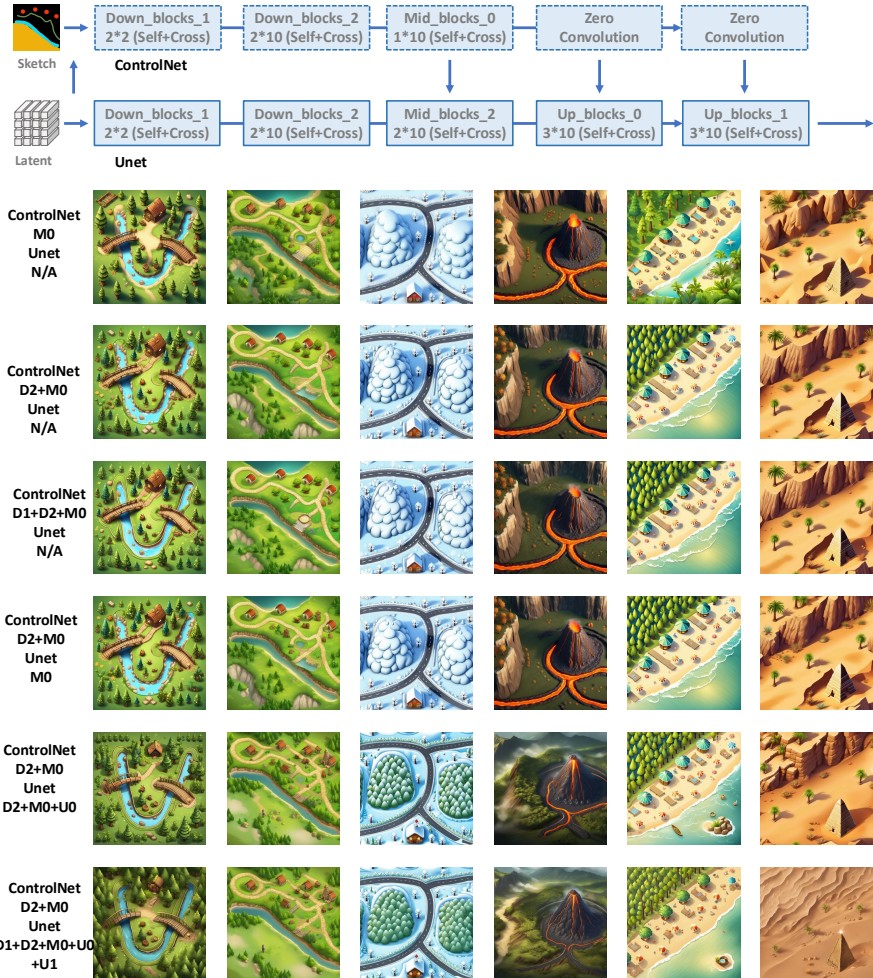

Figure 12: Qualitative comparison of Dense Tuning (DT) applied to different ControlNet layers. DT at `mid_block_0` yields coherent and well-controlled results, while extending DT to `down_block_2 + mid_block_0` further improves structural consistency. In contrast, applying DT to earlier layers (e.g., `down_block_1`) over-amplifies sketch contours and edges, leading to visually intrusive ControlNet outlines and degraded image naturalness. Applying DT to mid-level U-Net layers leads to marginal visual differences, whereas tuning peripheral U-Net layers introduces noticeable degrading changes. Prompts and Sketches are from Figures 8, 15, and 16.

consistently. Given the limited performance gains observed in Table 2 and the growing computational overhead shown in Table Y, restricting DT to `down_block_2` and `mid_block_0` yields the most favorable trade-off between controllability, efficiency, and overall visual quality.

Based on these qualitative and quantitative observations, we adopt `down_block_2 + mid_block_0` as the default DT configuration in ControlNet.

## C   Visualization of Game Scenes

Using a game scene as an example, we begin each prompt with 'Isometric view of a game scene' to generate controlled synthetic images for game settings. This helps maintain a consistent angle and style, ignoring any incoherent instance sketches that might appear in real-world scenes, thereby focusing on object placement and verifying text-image consistency. We generate all 50 complex scenes using hyperparameters identical to those used in the main results (Fig. 7 in the main paper), shown in Fig. 12 in the main paper and Fig. 16. The colored sketches are used solely to distinguish between different instances, and the colors used are arbitrary without class or semantic information. To validate this, we also use grayscale sketches as input, and the resulting images are nearly identical under the same random seed (two columns pointed by the red arrows in Fig. 16). Meanwhile, our approach is not limited to game scenes. We also test prompts without the fixed game scene phrase, resulting in more diverse angles and styles while maintaining the same quality in object placement and text-image consistency (One row pointed by the green arrows in Fig. 12 in the main paper.

## D   Visualization of Common Scenes

**Qualitative Common Scenes.** In the above experiments, we primarily validate the controllability of our method for multi-instance generation in game scenes. However, this does not imply that our approach is limited to game scenarios. To further verify its capabilities, we design three sets of diverse scenes: (1) four common simple scenes; (2) two indoor scenes; and (3) three scenes featuring instances of the same type but with different color attributes. Without changing any hyperparameters, generations are presented in Fig. 17. In common scenes, our method effectively mitigates instance overlap under ControlNet control, while in indoor scenes, it handles varied layouts well. For the challenging task of differentiating attributes within identical instances, our approach assigns distinct properties accurately. However, for uncommon attributes like generating a red cat, our method struggles due to limitations inherent in the original SDXL model.

## E   Metric of User Study

We conduct a user study on 50 scenes, each with 6 variants, generating 100 images per scene. A Gradio-based evaluation interface is designed, which randomly selects one image from 120 sets to create a sub-evaluation system, with images presented anonymously. 23 participants independently rate the images based on the following scale:

- **5**: All instances are accurately placed, and overall image quality is high.

- **4**: One instance is missing or misplaced, or All are placed with lower quality.

- **3**: Two or three instances are missing or misplaced, or placed with lower quality.

- **2**: Three or four instances are missing or misplaced, or placed with lower quality.

- **1**: Multiple instances are missing, with low overall quality.

This detailed rating system helps assess both the accuracy of instance placement and the quality of generated images, whether the generations are aligned with text prompts and sketch layouts.

## F   Transfer PB to Attend-and-Excite

To further validate the PB module, we integrated it into the Attend-and-Excite method (Chefer et al., 2023), based on attention tuning using the SD V1.4 model. The results are shown in Figure 13. Despite the

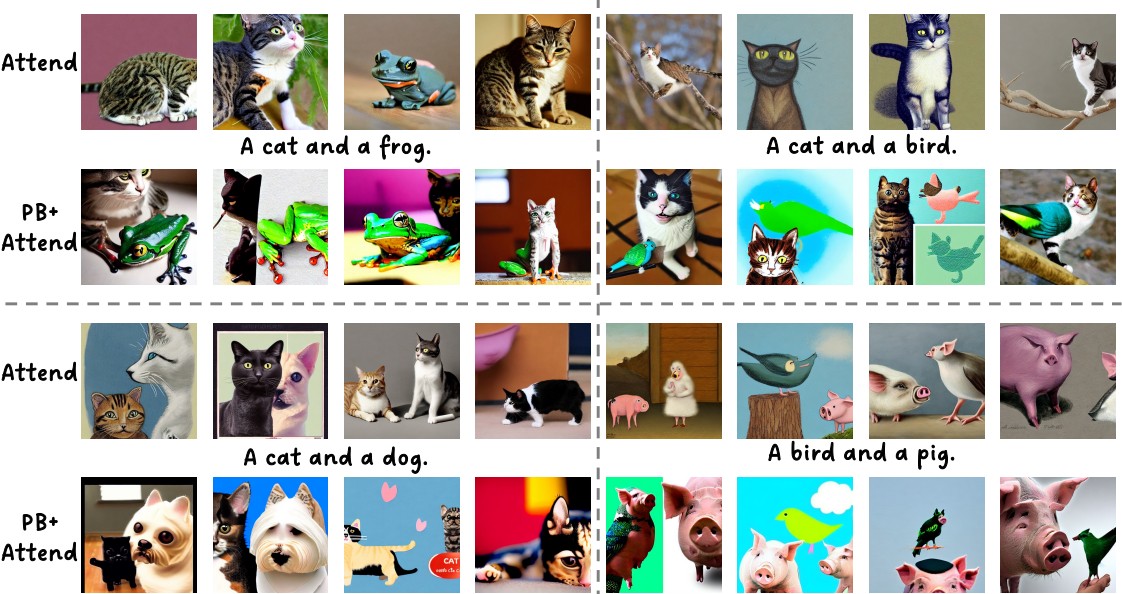

Figure 13: Visualizations for transferring PB to Attend-and-Excite (Chefer et al., 2023). In most cases, both instances are successfully generated. The frog-leg cat and the bird-wing pig further demonstrate the effectiveness since they lack the layouts to separate the instances spatially.

limitations of SD V1.4, the PB module effectively balances embedding strength between the two instances in scenarios without layout guidance, enhancing their representation. In most cases, both instances are successfully generated. However, in some cases, the attributes of the two objects become entangled, leading to artifacts such as a cat with frog legs or a pig with bird wings, due to the lack of spatial separation, which further demonstrates the effectiveness of the PB module.

## G   Transfer T$^3$-S2S to T2I-Adapter

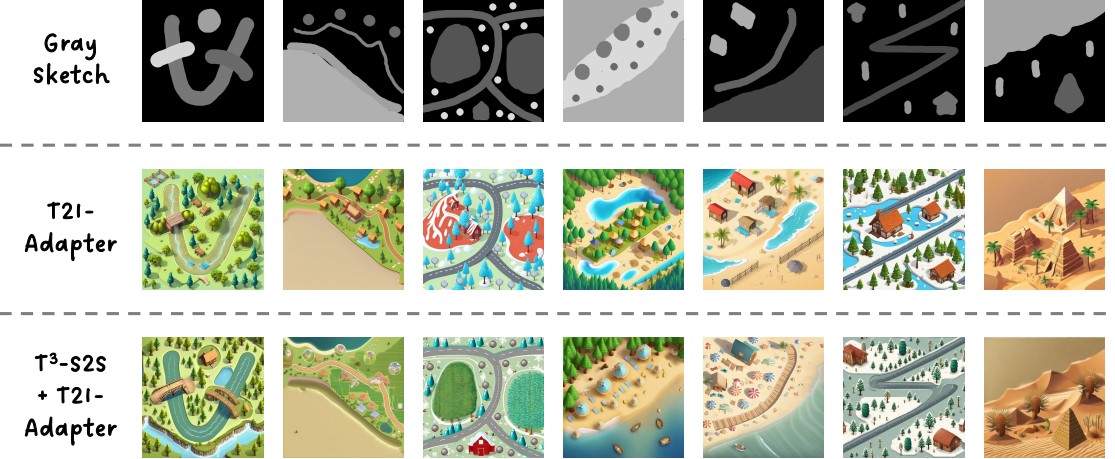

Figure 14: Visualizations for transferring T$^3$-S2S to T2I-Adapter (Mou et al., 2023). T$^3$-S2S effectively improves the T2I-Adapter's alignment with prompts and layouts in complex scenes, demonstrating its control capabilities across different models.

To validate the general applicability of our approach beyond the ControlNet model, we apply $T^3$-S2S to another controllable T2I-Adapter (Mou et al., 2023) model. Although the T2I-Adapter performs best with detailed sketches, we use grayscale sketches for quick validation, which contain less semantic information. We keep the PB and CP modules unchanged, while the DT module is integrated into the SDXL main channel, similar to CP, as it can not be placed in a separate branch like in ControlNet. We use the same prompts and sketches from the main results (Fig. 7 in the main paper) and Appendix C, with all other hyperparameters unchanged. The results are shown in Figure 14. $T^3$-S2S effectively improves the T2I-Adapter's alignment with prompts and layouts in complex scenes, demonstrating its control capabilities across different models. However, the generation quality still lags behind the ControlNet-based approach, indicating the need for parameter tuning specific to the T2I-Adapter's distribution and improved sketch inputs to align with the T2I-Adapter. Despite these limitations, the results show that $T^3$-S2S has promising generalizability and can effectively control both ControlNet and T2I-Adapter models.

## H   Perception Modeling for Structural Assembly

To bridge the transition between twin structural concept representations, i.e., isometric and terrain images, we follow sketch2scene (Xu et al., 2024) as our backend to build 3D game scenes from concept art generations. Structurally Specified Assembly module constructs full 3D natural scenes by segmenting foreground assets from isometric images, generating terrain meshes via tomography from terrain images, and applying sketch-based textures. This supports accurate terrain modeling, rich asset composition, and seamless integration with standard engines.

**Foreground Instances.** We extract 2D isometric images of individual objects (e.g., buildings, houses, trees) from the instance segmentation map via Segment Anything (Kirillov et al., 2023), which are later used for 3D retrieval or conditioning in the 3D generation. Assuming an isometric projection with a 45° yaw angle, we estimate object poses by applying homography warping (Hartley & Zisserman, 2003) and analyzing their bounding boxes. The warped instance segmentation provides the object footprint, while the depth information enables projection into 3D space.

**HeightMap Generation.** The isometric view offers sufficient coverage to recover both geometry and appearance from a single terrain image. We reconstruct a watertight terrain mesh from the image using Depth Anything (Yang et al., 2024) for depth estimation, followed by Poisson surface reconstruction. From the coarse depth, we obtain an isometric view depth map and compute the height as $h = d_{\max} - d$. The height is rotated into bird's-eye view (BEV) via homography warping (Hartley & Zisserman, 2003) to build the terrain mesh, and then the color-aligned mesh is segmented via Segment Anything (Kirillov et al., 2023) and Osprey (Yuan et al., 2023) into categories (grass, rock, mud, road, etc.). The categories map is used to retrieve standard texture tiles to ensure high-res appearance in close-up views. For water regions, we lower the terrain and insert a water asset to maintain realistic elevation.

**Procedural 3D Scene Synthesize.** Using semantic and geometric data, we procedurally generate and render 3D scenes via game engines (e.g., Unity, Unreal). We adopt Unity for its terrain, vegetation, and animation support. Heightmaps and splatmaps are mapped to Unity terrain. Vegetation is procedurally placed based on texture semantics (e.g., grass $\rightarrow$ flowers, rocks). Foreground objects are placed by retrieval (from Objaverse via CLIP) or generated using 2D-to-3D models (Hong et al., 2023; Wei et al., 2024; hyperhuman, 2024), ensuring style consistency with the reference.

3D scene generations following the sketch2scene (Xu et al., 2024) are shown in Figs. 19 and  20.

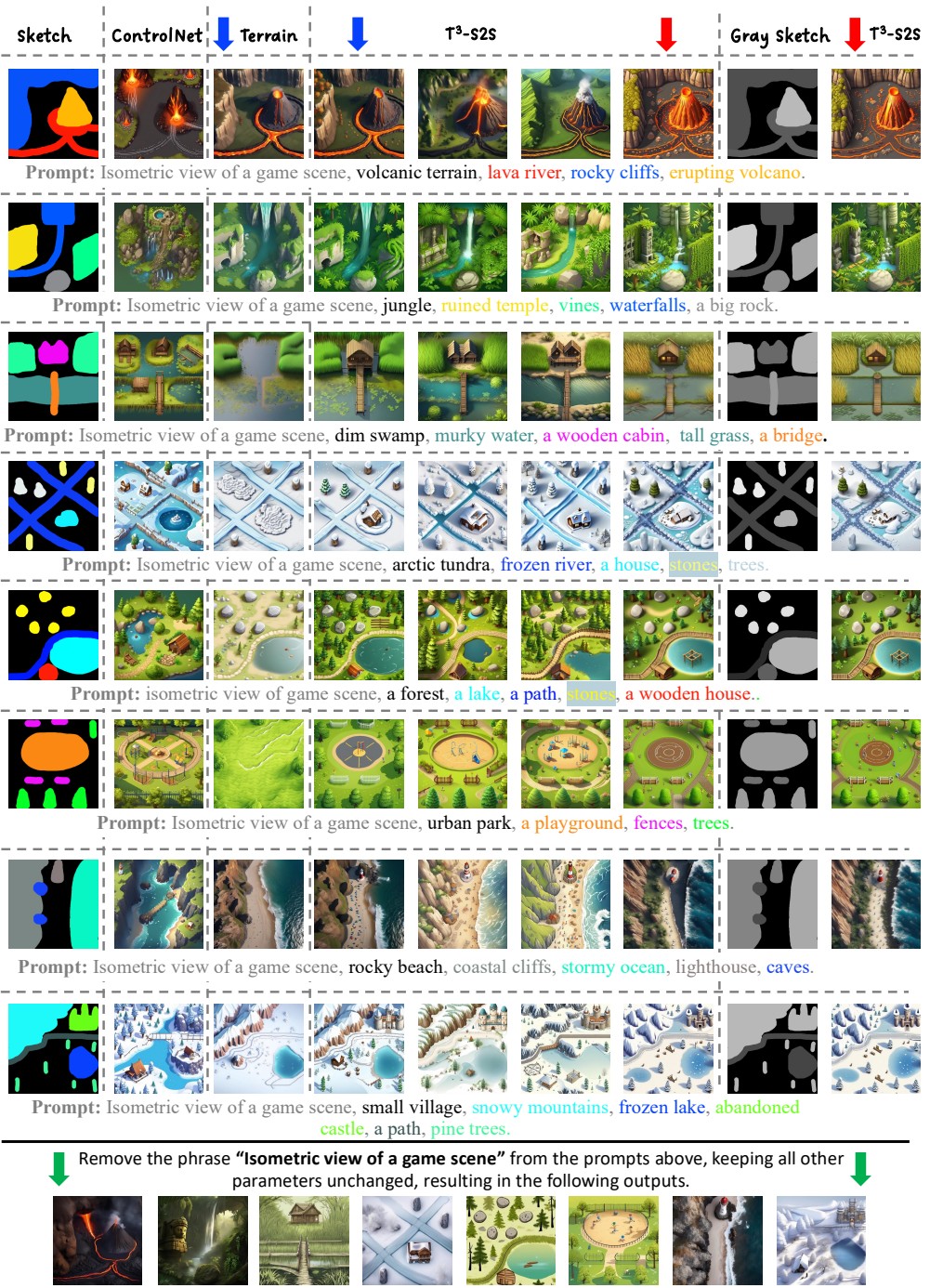

Figure 15: Example results from a subset of the 50 complex scene compositions tested using hyperparameters identical to those used in the main results (Fig. 8). (1) Two columns pointed by the blue arrows represent the structural concept representations using the prefix terrain prompts and isometric prompts under the same random seed. (2) Two columns pointed by the red arrows represent the generations using colored and grayscale sketches under the same random seed. (3) One row pointed by the green arrows indicates the generations without the fixed game scene phrase.

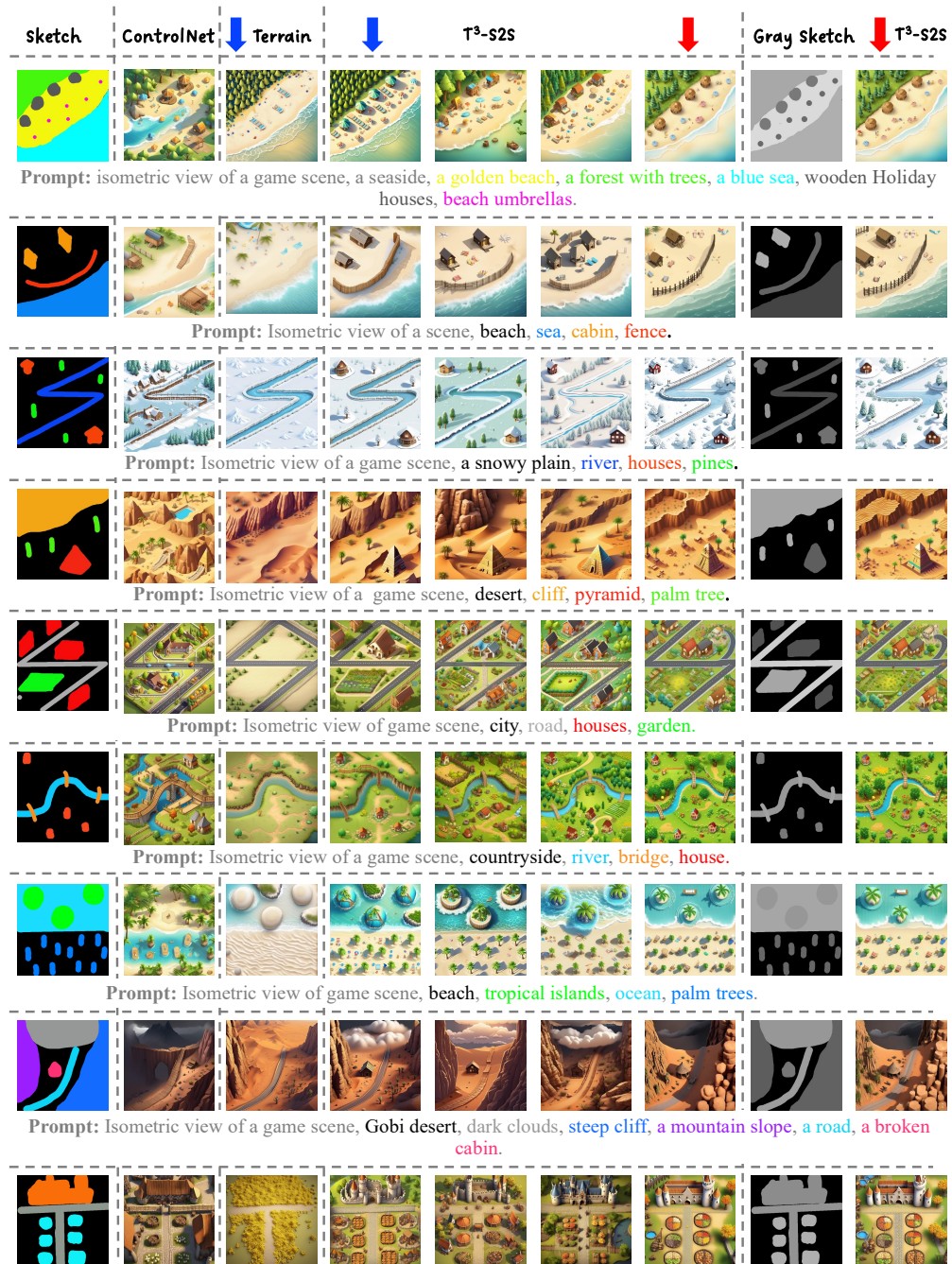

Figure 16: Example results from a subset of the 50 complex scene compositions tested using hyperparameters identical to those used in the main results (Fig. 7). (1) Two columns pointed by the blue arrows represent the structural concept representations using the prefix terrain prompts and isometric prompts under the same random seed. (1) Two columns pointed by the red arrows represent the generations using colored and grayscale sketches under the same random seed.

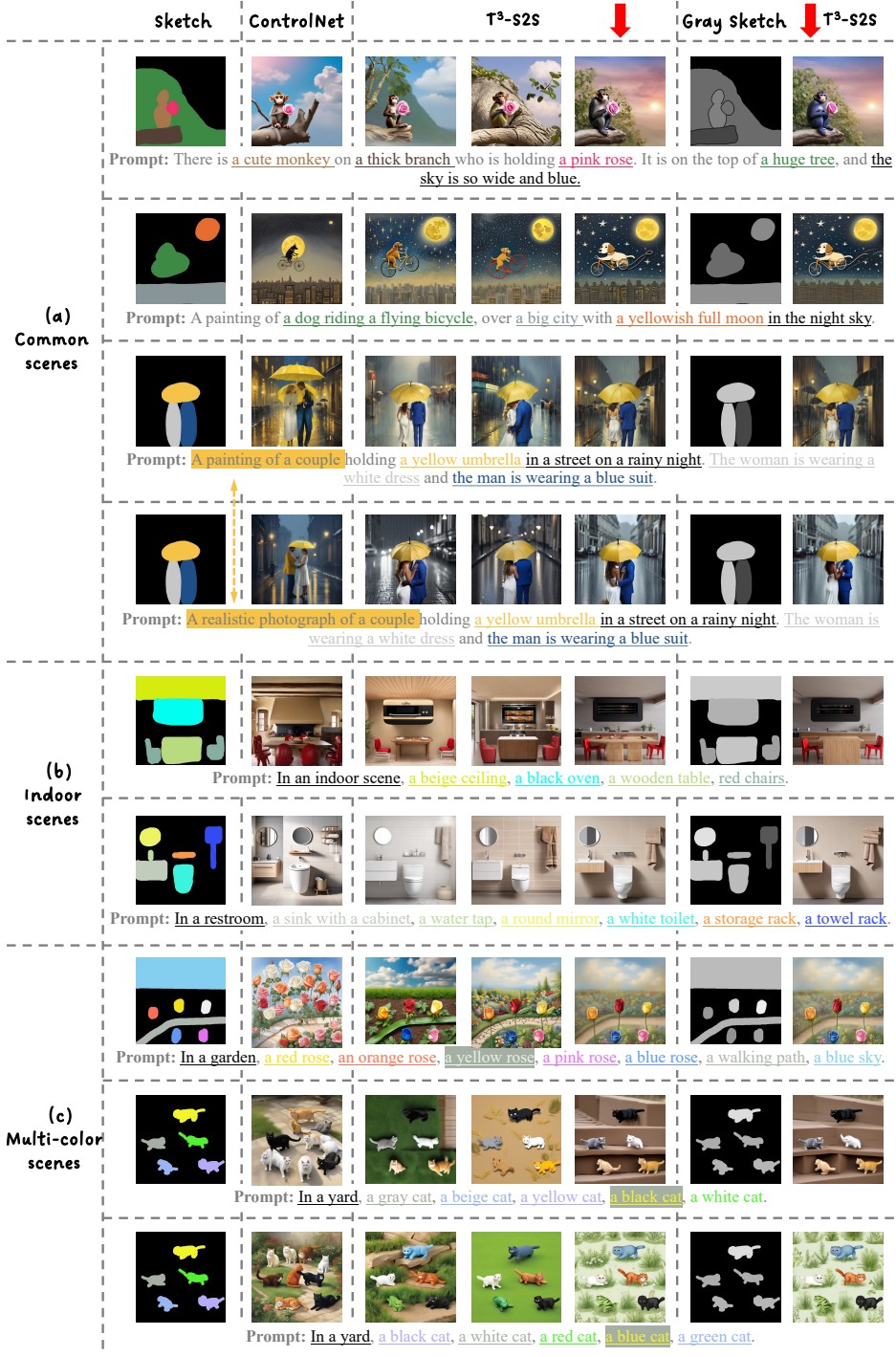

Figure 17: Examples of generated scenes across different settings. (a) Common simple scenes demonstrating effective instance representations under ControlNet control. (b) Indoor scenes showcasing robust handling of varied instance layouts. (c) Scenes with identical instances but different color attributes illustrate precise differentiation of properties.

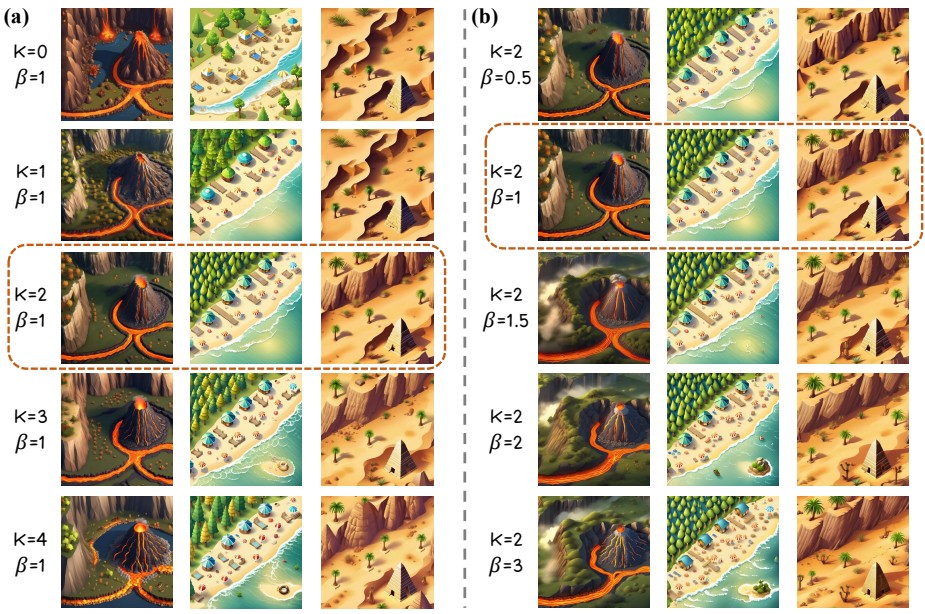

Figure 18: Another visual comparison of two hyper-parameters $K$ and $\beta$ suggests that setting $K = 2$ and $\beta = 1$ is a favorable choice. With $\beta = 1$, increasing $K$ initially improves results but eventually adds noise. Fixing $K = 2$, higher $\beta$ values yield stable and favorable outcomes.

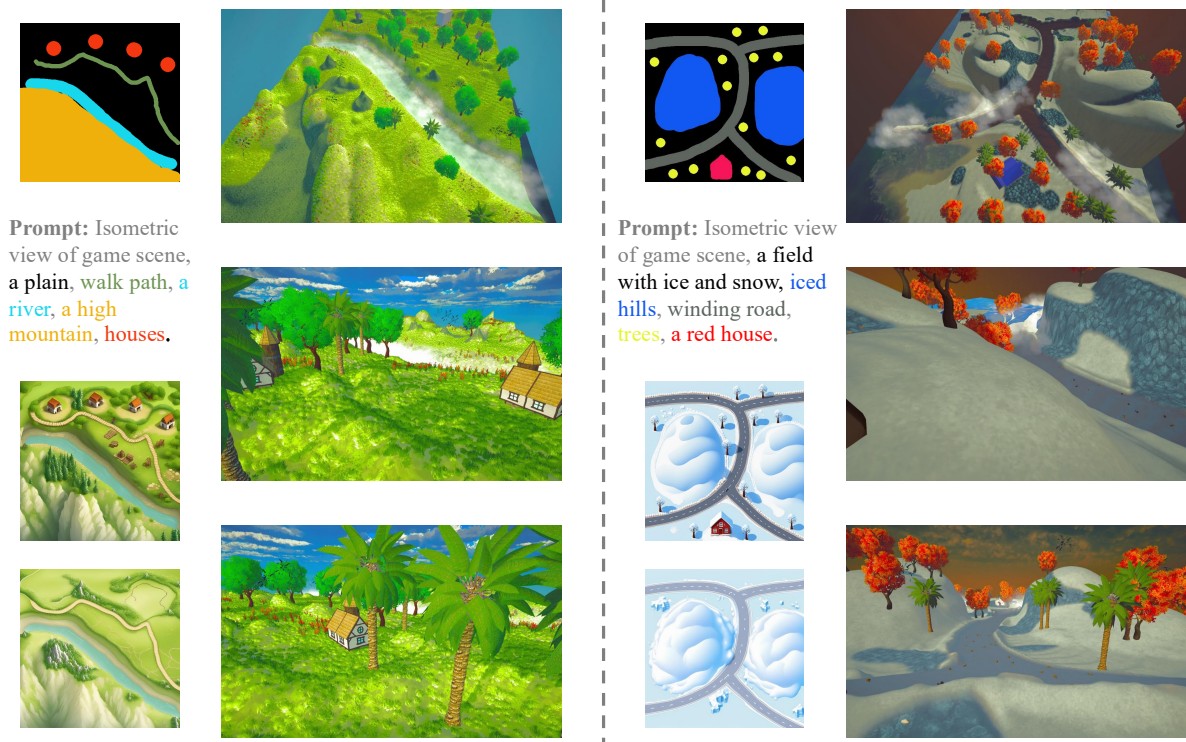

Figure 19: 3D scene generation results with two examples from Fig. 8. The 3D scene shows that terrain and objects can be generated as concept art, but improvements are needed in understanding concept art, particularly in segmentation and depth estimation, which may not be accurately represented in the synthesized data. Additional 3D scenes are available in the Appendix and the supplementary video materials.

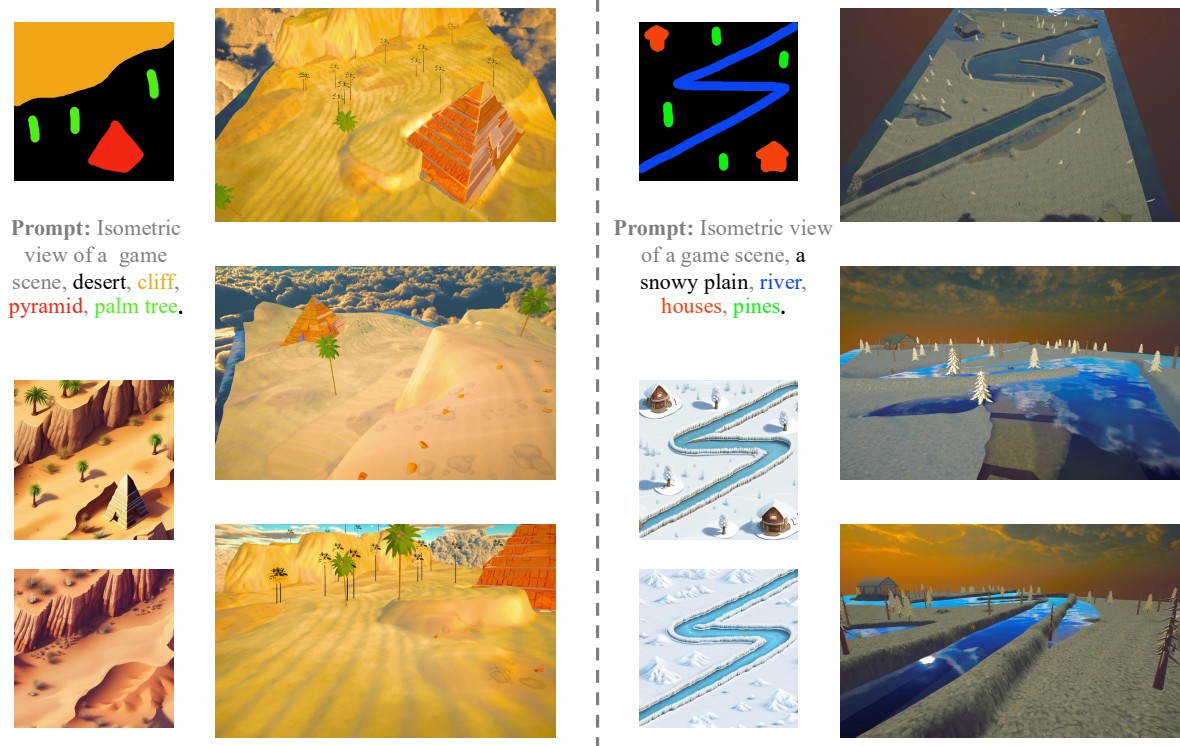

Figure 20: 3D scene generation results with two more examples from Fig. 16. The 3D scene shows that terrain and objects can be generated as concept art, but improvements are needed in understanding concept art, particularly in segmentation and depth estimation, which may not be accurately represented in the synthesized data.

