# OpenReview forum: "T$^3$-S2S: Training-free Triplet Tuning for Sketch to Scene Synthesis in Controllable Concept Art Generation"
_TMLR — Accepted by TMLR_

### Review · Reviewer_wShr · 2025-09-08

**Summary Of Contributions:**

The paper addresses a critical challenge in AI-generated content creation: improving image/video generation without fine-tuning by enhancing the generation of items mentioned in sketches and prompts. The paper identifies key shortcomings in existing solutions, particularly the problem of forgetting low-energy key items from text and sketches, and proposes a solution to increase the energy of these low-energy keys during generation.

**Audience:**

Yes

**Audience Explanation:**

Yes. The findings would be of clear interest to TMLR’s audience, particularly those working on controllable text-to-image generation, diffusion models, and cross-attention mechanisms. The paper addresses a well-motivated challenge—multi-instance generation—and proposes a modular, training-free solution with strong theoretical backing and practical utility. Researchers and practitioners in machine learning, computer vision, and generative modeling would benefit from both the methodological insights and the empirical results.

**Broader Impact Concerns:**

.

**Claims And Evidence:**

Yes

**Claims Explanation:**

This paper makes a strong contribution to controllable image generation through a well-defined problem statement, rigorous theoretical analysis of cross-attention mechanisms, and an innovative three-module solution (Prompt Balance, Characteristics Priority, and Dense Tuning). The proposed approach is training-free, practical, and cost-effective, while maintaining compatibility with ControlNet’s sketch-following ability. The authors support their claims with solid experimental validation, including CLIP scores, user studies, ablations, and transfer tests to related models.

**Requested Changes:**

The main requested change is to provide a clear justification for applying Dense Tuning only at “down_block_2” and “mid_block_0” in ControlNet. To strengthen the paper, the authors should include empirical comparisons of different layer combinations, an analysis of the ControlNet feature hierarchy to explain why mid-level features are most effective, and a cost-benefit evaluation covering memory usage, inference time, and performance trade-offs. In addition, providing guidelines on how to adapt layer selection for other architectures would greatly improve reproducibility and generalizability.

---

> ### Author Response · Authors · 2025-12-22
> **Response**
>
> Dear reviewer,
>
> We thank the reviewer for the insightful suggestion regarding the justification of applying Dense Tuning (DT) only at `down_block_2` and `mid_block_0` in ControlNet. We have substantially revised the manuscript with a dedicated section in **Appendix B** (“Details of Dense Tuning”) that provides empirical, architectural, and efficiency-based justification, as summarized below.
>
> **(1) Empirical comparison of different layer combinations.**
> We systematically evaluate DT applied to different ControlNet depths within the same PB+CT framework. As shown in the qualitative comparison (Fig. 12), DT applied at `mid_block_0` already produces coherent and well-controlled results, while extending DT to `down_block_2 + mid_block_0` further improves structural consistency. In contrast, including earlier layers such as `down_block_1` leads to overly strong contour and edge responses, resulting in visually intrusive outlines and degraded image naturalness. These results demonstrate that tuning early layers is suboptimal, while mid-level layers offer the most effective control.
>
> **(2) Analysis of ControlNet feature hierarchy.**
> We follow empirical observations [1,2,3] in Diffusion models, following established layer- and step-wise observations in diffusion models. Early (peripheral) layers primarily encode low-level geometric cues and boundary information, whereas more central layers shape object contours and structural layouts. This hierarchy explains why dense modulation at early layers tends to over-amplify sketch boundaries, while mid-level layers provide more semantically meaningful and balanced structural guidance. This analysis is consistent with the architectural design of diffusion UNets and ControlNet, where semantic structure progressively emerges at intermediate depths.
>
> **(3) Cost–benefit evaluation.**
> We further provide a quantitative cost–benefit analysis covering GPU memory usage and inference latency (Table 3). As DT is activated in additional layers, both memory consumption and inference time increase consistently, while performance gains remain marginal. This demonstrates diminishing returns when extending DT beyond a minimal set of mid-level layers. Consequently, restricting DT to `down_block_2` and `mid_block_0` achieves the most favorable trade-off between controllability, efficiency, and visual quality.
>
> **(4) Guidelines for adaptation to other architectures.**
> Based on these observations, we provide a practical guideline for layer selection: DT should be applied to mid-level blocks where conditioning signals interact with structural and semantic representations, while avoiding very early layers that encode low-level geometry and late layers that are highly task-specific. This guideline is architecture-agnostic and can be readily adapted to other ControlNet or UNet-based diffusion backbones.
>
> Overall, these additions clarify that DT functions as a modulation mechanism rather than the primary performance driver (which is provided by PB and CT), and that `down_block_2 + mid_block_0` represents a principled and efficiency-aware design choice.
>
> ***
> [1] Voynov, Andrey, et al. "p+: Extended textual conditioning in text-to-image generation."
>
> [2] Sun, Zhenhong, et al. "Eggen: Image generation with multi-entity prior learning through entity guidance."
>
> [3] Wang, Junyan, et al. "Towards effective usage of human-centric priors in diffusion models for text-based human image generation."
> ***
>
> Best regards,
>
> Authors

---

> > ### Comment · Reviewer_wShr · 2025-12-23
> > **Looks good, thank you for adding additional info. I don't have any further questions.**
> >
> > .

---

### Review · Reviewer_pQcf · 2025-10-06

**Summary Of Contributions:**

A training free method to synthesize a scene based on a sketch consist of three modules: Prompt Balance (PB), Characteristics Priority (CP), and Dense Tuning (DT). PB finds instance keywords in the prompt, replaces multi-word phrases with single-token embeddings, and rescales their norms to match the End-of-Text token, keeping the prompt energy balanced across instances. CP applies a per-channel Top-K selection on value activations to highlight instance-relevant features within sketch regions. DT modifies ControlNet with a lightweight Dense Diffusion step that sharpens attention at specific blocks without retraining. The method also uses a dual-branch framework to generate both isometric and terrain views, leading to more coherent and controllable terrain layouts.

Strength:

1. The method is training free require no additional data
2. good theoretical justifications
3. comprehensive ablations

Weakness:

1. limited analysis of failure cases
2. quantitative evaluation not super impressive (I am not by any means an expert in this field so I don't know what significance do those number represents)
3. the quantitative results only has 50 examples, and is not a publicly open dataset, which can cause some doubts.

**Additional Comments:**

This area is totally out of my expertise so I am especially unconfident about my judgement.

**Audience:**

Yes

**Audience Explanation:**

yes, for people working on controllable generation

**Claims And Evidence:**

Yes

**Claims Explanation:**

Yes, figure 10 and table 1 shows the quantitative results while many other figures present qualitative results, which looks convincing.

**Requested Changes:**

The quantitative results is not solid enough. Could you expand you analysis to a higher scale for better statistical significance? Also would be great if there are some disclosure in how the eval set are constructed.

---

> ### Author Response · Authors · 2025-12-22
> **Response**
>
> Dear reviewer,
>
> We deeply appreciate your constructive comments for improving our paper. We would be incredibly grateful for your continued support in reviewing our response.
>
>
> ***
> Q1: Limited analysis of failure cases.
>
> A1: We thank the reviewer for this constructive suggestion. In the revised manuscript, we have included a detailed discussion on limitations and failure cases in Appendix A. Our analysis primarily focuses on two aspects: 1) Texture generation challenge: We discuss the occasional attribute confusion (e.g., color mismatches) caused by the limited expressivity of CLIP representations and the prioritization of instance-level tokens over adjectival descriptors. 2) Large scene challenge: We analyze the difficulty in controlling minute instances in expansive scenes (>$4096^2$), which is attributed to the intrinsic $32\times$ downsampling bottleneck of the SDXL backbone.
>
> ***
>
> Q2: Higher scale for better statistical significance.
>
> A2: We appreciate the suggestion to strengthen our quantitative evaluation. We have revised Section 5.2 to incorporate the large-scale T2I-CompBench results.
>
> * Scale & Significance: We expanded the analysis to the T2I-CompBench benchmark, which includes 300 prompts per metric (Color, 2D Spatial, Numeracy) and generates 10 images per prompt. This large-scale evaluation confirms our method's significant advantages, showing a 9.5\% improvement in Color and 15.7% in Spatial metrics over baselines.
>
> * Dataset Construction: We have explicitly disclosed the construction details in the revised text. To adapt the text-only benchmark for our layout-driven method, we employed GPT-4V to generate instance sketches. For the Numeracy test, we specifically filtered for challenging scenarios containing 1-3 small instances with minimal resolution constraints (ensuring feature viability), where our method demonstrated a distinct 16.2\% gain over ControlNet. For the customized 50 complex scenes, we provided 20 examples in Figures 16 and 17, which were hand-drawn one by one.
>
> ***
>
> Best regards,
>
> Authors

---

### Review · Reviewer_yNzH · 2025-12-08

**Summary Of Contributions:**

The paper introduces *Training-free Triplet Tuning for Sketch-to-Scene (T3-S2S)*, a novel method to improve the controllability and accuracy of multi-instance scene generation from sketches.

 **Key Contributions:**
- Theoretical Insights: The authors investigate cross-attention mechanisms and identify problems with prompt energy imbalance, leading to poor instance representation.
- T3-S2S Framework: Combines three modules:
  - Prompt Balance (PB): Adjusts keyword energy to improve instance token competition.
  - Characteristics Priority (CP): Enhances instance prominence using TopK value matrices.
  - Dense Tuning (DT): Refines attention maps for better instance contours.
- Feature Sharing: A dual-branch prompt synthesis for creating isometric and terrain-view representations.
- Empirical Validation: Demonstrates that T3-S2S outperforms existing models like ControlNet in generating detailed and accurate multi-instance scenes.

 **Strengths**:
- Training-free: No additional data required, adaptable, and cost-effective.
- Improved instance generation: Addresses issues of missing small and rare instances in previous methods.
- Clear experimental results: Empirical evidence supports the effectiveness of the method.

 **Weaknesses**:
- Limited scalability: Struggles with large, complex scenes that require detailed inter-instance relationships.
- Lack of direct comparison: No detailed comparison with cutting-edge models for large-scale scene generation.

**Audience:**

Yes

**Audience Explanation:**

- The paper contributes to improving sketch-to-scene generation.
- Game developers, filmmakers, and content creators can benefit from the advancements in automating concept art generation.
- The training-free nature of the approach is relevant to those working on scalable AI systems.

**Broader Impact Concerns:**

- Bias in generation: While the method is training-free, biases could still emerge from base models like ControlNet. The paper should discuss potential fairness and bias issues in content generation.

**Claims And Evidence:**

Yes

**Claims Explanation:**

- Clear visual comparisons with baseline models (e.g., ControlNet) showing improved handling of smaller and complex instances.
- CLIP-Score evaluations for global image quality, instance representation, and background accuracy show significant improvements.

**Requested Changes:**

1. **Clarify Handling of Large-Scale Scenes**:
   - The paper mentions challenges with generating large-scale scenes, but this section could be expanded to explain *why* the method struggles with large scenes, particularly in maintaining complex relationships between multiple objects.
   - The authors should provide insights into how the current model's limitations impact the synthesis of large, intricate scenes (e.g., scenes with overlapping objects or large amounts of detail) and potential improvements or directions for overcoming these challenges in future work.

2. **More Detailed Hyperparameter Tuning Discussion**:
   - The tuning of hyperparameters like **TopK** and **β** plays a crucial role in the performance of the model. The paper should include more explicit guidance on:
     - How **TopK** and **β** values were selected for the experiments (e.g., grid search, manual selection).
     - The impact of different values on the model’s performance across various types of scenes (e.g., small scenes vs. large scenes, or simple vs. complex prompts).
     - A clearer discussion of how these parameters could be adjusted in practical applications to improve scene generation for different types of inputs.

3. **Comparison with Cutting-Edge Methods for Large-Scale Scenes**:
   - While the paper demonstrates improvements over existing methods, it lacks a direct comparison with recent state-of-the-art models for large-scale scene generation, especially in cases that involve complex inter-object relationships.
   - It would strengthen the paper to add an explicit comparison of T3-S2S with recent large-scale scene generation methods to show its relative performance in generating high-fidelity multi-instance scenes.

4. **Further Explanation of Dense Tuning Modifications**:
   - The **Dense Tuning (DT)** module is key to improving performance, but the paper could elaborate more on why dense tuning at certain layers ("down_block_2" and "mid_block_0") yields the best results. A clearer explanation of why these layers were specifically chosen and how their tuning impacts performance would be beneficial.
   - A deeper analysis of potential drawbacks or trade-offs when applying dense tuning to these layers could help readers better understand its limits and provide guidance for future applications.

---

> ### Author Response · Authors · 2025-12-22
> **Response-Part1**
>
> Dear reviewer,
>
> We deeply appreciate your constructive comments for improving our paper. We would be incredibly grateful for your continued support in reviewing our response.
>
>
> ***
> Q1: Clarify Handling of Large-Scale Scenes.
>
> A1: We thank the reviewer for this insightful comment regarding large-scale scene generation and comparisons.
>
> **1. Scope Clarification & Comparison**
>
> We respectfully clarify that our primary contribution lies in controllable layout-to-image generation (ensuring precise alignment with user sketches), which is fundamentally distinct from unbounded large-scale synthesis (e.g., methods like ScaleCrafter). Specialized large-scale methods primarily focus on resolution upscaling or infinite tiling strategies and typically do not support strict, instance-level layout control defined by users. Consequently, a direct comparison would be inequitable due to these differing task formulations. Instead, we benchmark against state-of-the-art controllable models (e.g., ControlNet, DenseDiffusion), as they represent the relevant baseline for evaluating the model's ability to handle complex inter-object relationships under explicit layout constraints.
>
> **2. Mechanism Analysis & Future Work (Appendix A)**:
>
> Addressing the "why" and ``how" of large-scale failures, we have expanded Appendix A to provide a deeper architectural analysis:
>
>  * The ``Why" (32$\times$ Limit): We explain that the difficulty in generating expansive scenes (e.g., $4096^2$) stems from the intrinsic 32$\times$ downsampling bottleneck of the SDXL architecture. The model compresses a $1024^2$ input into a $32^2$ feature map, creating a "structural floor." Object interactions or details smaller than this threshold suffer from spatial aliasing, making precise control theoretically intractable regardless of the output resolution.
>
>  * The Future Work: We added future directions in the revised text, suggesting that coarse-to-fine cascade architectures or spatial-aware latent upsampling are necessary strategies to bypass this fixed downsampling limit and effectively resolve intricate relationships in large-scale maps.
>
> ***
>
> Q2: More Detailed Hyperparameter Tuning Discussion.
>
> A2:We thank the reviewer for this insightful comment regarding hyperparameter selection. In the revised manuscript, we have completely rewritten Section 5.4 and added a new Figure 18 to provide a rigorous analysis.
>
> **1. Selection Methodology**: We clarify that our parameters were not chosen heuristically but through distinct analytical processes:
>
> * $K$ (Statistical Analysis): As shown in the histogram in Figure 10, we statistically analyzed the distribution of extreme values within the attention matrices. We observed that the probability of token significance drops sharply after the second extremum. Thus, $K=2$ is derived from data statistics to maximize signal while minimizing noise.
>
> * $\beta$ (Grid Search): We conducted an empirical grid search (visualized in Figures 11 and 18 (right)) to find the optimal trade-off between feature enhancement and structural coherence.
>
> **2. Robustness and Fixed Settings**: Crucially, we emphasize that these hyperparameters are fixed ($K=2, \beta=1$) across all our experiments. Unlike methods that might require per-image tuning, our analysis proves that this configuration provides a robust baseline applicable to diverse scenarios (from simple objects to complex scenes) without the need for manual adjustment.
>
> ***

---

> > ### Author Response · Authors · 2025-12-22
> > **Response-Part2**
> >
> > ***
> >
> > Q3: Further Explanation of Dense Tuning Modifications:
> >
> > A3: We thank the reviewer for the insightful suggestion regarding the justification of applying Dense Tuning (DT) only at `down_block_2` and `mid_block_0` in ControlNet. We have substantially revised the manuscript with a dedicated section in **Appendix B** (“Details of Dense Tuning”) that provides empirical, architectural, and efficiency-based justification, as summarized below.
> >
> > **(1) Empirical comparison of different layer combinations.**
> > We systematically evaluate DT applied to different ControlNet depths within the same PB+CT framework. As shown in the qualitative comparison (Fig. 12), DT applied at `mid_block_0` already produces coherent and well-controlled results, while extending DT to `down_block_2 + mid_block_0` further improves structural consistency. In contrast, including earlier layers such as `down_block_1` leads to overly strong contour and edge responses, resulting in visually intrusive outlines and degraded image naturalness. These results demonstrate that tuning early layers is suboptimal, while mid-level layers offer the most effective control.
> >
> > **(2) Analysis of ControlNet feature hierarchy.**
> > We follow empirical observations [1,2,3] in Diffusion models, following established layer- and step-wise observations in diffusion models. Early (peripheral) layers primarily encode low-level geometric cues and boundary information, whereas more central layers shape object contours and structural layouts. This hierarchy explains why dense modulation at early layers tends to over-amplify sketch boundaries, while mid-level layers provide more semantically meaningful and balanced structural guidance. This analysis is consistent with the architectural design of diffusion UNets and ControlNet, where semantic structure progressively emerges at intermediate depths.
> >
> > **(3) Cost–benefit evaluation.**
> > We further provide a quantitative cost–benefit analysis covering GPU memory usage and inference latency (Table 3). As DT is activated in additional layers, both memory consumption and inference time increase consistently, while performance gains remain marginal. This demonstrates diminishing returns when extending DT beyond a minimal set of mid-level layers. Consequently, restricting DT to `down_block_2` and `mid_block_0` achieves the most favorable trade-off between controllability, efficiency, and visual quality.
> >
> > **(4) Guidelines for adaptation to other architectures.**
> > Based on these observations, we provide a practical guideline for layer selection: DT should be applied to mid-level blocks where conditioning signals interact with structural and semantic representations, while avoiding very early layers that encode low-level geometry and late layers that are highly task-specific. This guideline is architecture-agnostic and can be readily adapted to other ControlNet or UNet-based diffusion backbones.
> >
> > Overall, these additions clarify that DT functions as a modulation mechanism rather than the primary performance driver (which is provided by PB and CT), and that `down_block_2 + mid_block_0` represents a principled and efficiency-aware design choice.
> >
> > ***
> > [1] Voynov, Andrey, et al. "p+: Extended textual conditioning in text-to-image generation."
> >
> > [2] Sun, Zhenhong, et al. "Eggen: Image generation with multi-entity prior learning through entity guidance."
> >
> > [3] Wang, Junyan, et al. "Towards effective usage of human-centric priors in diffusion models for text-based human image generation."
> > ***
> >
> > ***
> >
> > Q4: Bias in generation.
> >
> > A4: We thank the reviewer for raising this crucial ethical consideration. We acknowledge that despite being training-free, our framework relies on pre-trained backbones (SDXL) and adapters (ControlNet), which inevitably introduce inherited biases.
> >
> > * Inherited Biases: We clarify that our model reflects the data distributions of the base models, potentially propagating societal stereotypes regarding gender, race, or cultural artifacts.
> >
> > * Mitigation via Control: On a positive note, we argue that our core contribution—precise layout control—offers a tool to counteract structural biases. By allowing users to explicitly define the position and size of instances, our method prevents the model from defaulting to stereotypical compositions (e.g., placing certain groups only in the background), thus empowering users to create more diverse and inclusive scenes.
> >
> > ***
> >
> > Best regards,
> >
> > Authors

---

### Author Response · Authors · 2025-12-22
**General Resonse**

Dear AE and Reviewers,

Thank you for coordinating the reviewers’ thoughtful evaluations and for forwarding the detailed feedback. We sincerely appreciate the time and efforts devoted to assessing our work.

In response to the reviewers’ comments, we have carefully revised the manuscript. The major changes are summarized below (highlighted in a blue font):

1. We added quantitative results on large-scale T2I-CompBench in Section 5.2 to provide a more comprehensive and objective evaluation of the proposed method.

2. We expanded the Top-(K) analysis in Section 5.4 to better justify the choice of hyperparameters. In addition, we included a detailed discussion of the newly added Figure 18 to further support our observations.

3. We substantially revised Appendix A to provide a clearer and more thorough analysis of the limitations of our approach.

4. We supplemented Appendix B with additional implementation details on Dense Tuning, and added Table 3 and Figure 12 to improve clarity and reproducibility.

We believe these revisions have significantly improved the clarity, completeness, and rigor of the paper. Thank you again for your time and dedication in handling our submission.

Best regards,
Authors

---

### Decision · Action_Editor_uLrk · 2026-01-20

**Recommendation:** Accept as is

**Audience:**

Yes

**Audience Explanation:**

The work is directly relevant to researchers and practitioners interested in controllable diffusion models, cross-attention mechanisms, and layout- or sketch-conditioned generation

**Claims And Evidence:**

Yes

**Claims Explanation:**

The paper’s central claims are improving controllable sketch-to-scene generation without training via Prompt Balance, Characteristics Priority, and Dense Tuning. These claims are well supported by a combination of clear mechanism analysis, systematic ablations, and both qualitative and quantitative evaluations. Visual comparisons demonstrate improved multi-instance alignment and reduced instance omission, while quantitative results (including CLIP-based metrics and large-scale T2I-CompBench evaluations added in revision) corroborate these gains. The revised manuscript also provides principled justifications for design choices (e.g., layer selection for Dense Tuning) and discusses limitations, which together make the evidence convincing and transparent.

---

> ### Author Response · Authors · 2026-01-26
>
> Dear Action Editor Prof. Fu,
>
> Thank you very much for your kind decision and support. We sincerely appreciate your handling of our submission, and we are also grateful to the reviewers for their constructive comments and valuable suggestions throughout the review process.
>
> Best regards,
>
> Authors